# MEDQUANBENCH: QUANTIZATION-AWARE ANALYSIS FOR EFFICIENT MEDICAL IMAGING MODELS

## ABSTRACT

Quantization is a crucial technology for facilitating the deployment of medical AI models, especially on 3D radiological data. However, existing studies often lack comprehensive evaluations across diverse architectures, modalities, and quantization techniques, which limits our understanding of the real-world trade-offs among applicability, efficiency, and performance. In this work, we introduce MedQuanBench, a large-scale and diverse benchmark designed to rigorously evaluate quantization techniques for 3D medical imaging models. Our benchmark spans a wide range of modern architectures (e.g., CNNs and Transformers). We systematically evaluate representative post-training quantization strategies across model scales and dataset sizes. Additionally, we perform detailed sensitivity analyses to identify which model components are most vulnerable to quantization, including layer-wise degradation and activation distribution shifts. Our results show that 8-bit quantization consistently preserves segmentation accuracy across diverse architectures, making it a reliable choice for deployment. Furthermore, with appropriate configuration, such as selecting proper quantization granularity based on the model structure, 4-bit precision can also achieve near-lossless performance. These results show MedQuanBench as a foundation for optimizing quantization and guiding the development of deployment-ready, low-bit medical imaging models.

## 1 INTRODUCTION

Medical foundation models have demonstrated remarkable performance across various clinical tasks such as segmentation (Isensee et al., 2021; He et al., 2021), classification (Li et al., 2023a), and generation (Guo et al., 2025). However, growing model sizes, driven by performance demands, combined with the expanding medical imaging datasets (Wasserthal et al., 2022; Qu et al., 2023), create significant computational challenges. In clinical practice, hardware resources are typically constrained, and the computational demands lead to worse inference latency and memory consumption (Tang et al., 2022; Gao et al., 2022), which prohibits real-world deployment. Quantization provides a promising approach (Frantar et al., 2023; Xiao et al., 2023) to reduce model complexity and computational costs by representing model weights and activations with fewer bits (Dettmers et al., 2022; Lin et al., 2024), typically converting high-precision (e.g., 32-bit floating-point) parameters into lower-precision formats (e.g., 4-8 bit integers). This optimization significantly decreases memory consumption, accelerates inference speed, and enhances hardware utilization without modification of training configuration or architecture design. While large language models (LLMs) and computer vision (Li et al., 2023b; Shang et al., 2023) have been extensively studied and exploited with low-bit quantization, medical imaging models still rely on high-precision formats (e.g., FP16 and FP32) (Huang et al., 2023b; Roy et al., 2023), showing a critical gap in exploring low-bit efficiency for the medical domain. A systematic benchmark is thus essential to explore optimal quantization techniques that can quantify trade-offs between bit-width, inference speed, memory consumption, and accuracy.

Post-training quantization (PTQ) has achieved success in large language models and natural image tasks (Xiao et al., 2023; Zhang & Chung, 2024; Li et al., 2024a; Ashkboos et al., 2024). However, adoption in medical imaging remains limited, motivating five unresolved challenges. **First**, current SOTA quantization techniques, such as activation smoothing (Xiao et al., 2023), singular value decomposition (SVD) (Li et al., 2024a), and rotation (Ashkboos et al., 2024), are primarily designed and validated on transformer-based models or linear layers. While Fig. 1(a) indicates convolution blocks dominate compute in medical models. The effectiveness of these techniques on convolutional

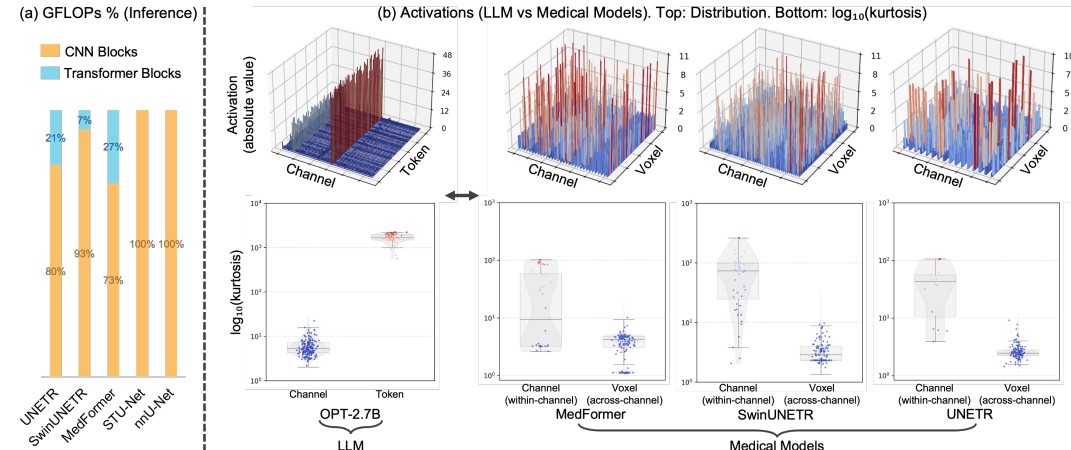

Figure 1: **Status of medical models.** *(a) Inference GFLOPs composition of representative medical imaging architectures.* Most compute sits in convolution blocks, while transformer blocks take a smaller share. This shows CNNs still dominate practical medical models. *(b) Activation behavior in an LLM (OPT-2.7B) and in medical models.* Top row shows activation distributions. For medical models, spatial voxels are flattened along the horizontal axis. Bottom row shows $\log_{10}$(kurtosis) Each dot is one unit: a channel or token for the LLM, and a channel or voxel for the medical models. Color intensity encodes magnitude, **blue** is lower and **red** is higher. Unlike LLMs, where outliers concentrate in a small set of channels and persist across tokens. Medical models exhibit the opposite pattern: outliers are spatially sparse within channels.

architectures (Liu et al., 2023; Yu et al., 2023), on diverse kernel shapes (e.g., anisotropic 3D convolutions) (Chen et al., 2021; Gao et al., 2022), and on non-GEMM operations (Gu et al., 2024) (e.g., depth-wise convolution) remains unclear. **Second**, medical imaging datasets exhibit unique activation patterns (Landman et al., 2015; Li et al., 2024b), typically characterized by spatially localized activation outliers rather than uniform channel-wise distributions (Wasserthal et al., 2022). Fig. 1(b) presents activation distributions (top) and $\log_{10}$ kurtosis (bottom) for an LLM (OPT-2.7B) and medical models (MedFormer, SwinUNETR, UNETR). In medical models, outliers are present across many channels with uneven intensity and are spatially localized. This is the opposite of LLMs, where outliers concentrate in a small set of channels and persist across tokens. These differences further indicate that PTQ methods effective on LLMs require additional validation in the medical domain. **Third**, existing evaluations (Bassi et al., 2024; Huang et al., 2023a) focused on accuracy comparisons across model architectures, are ignoring hardware deployment constraints such as memory footprint, latency on edge devices. **Fourth**, as LLMs and vision transformers tolerate aggressive quantization (INT4/INT8) (Sui et al., 2024; Xu et al., 2023), medical models face stricter accuracy demands. Currently, no studies have systematically analyzed the trade-offs between bit-width, task complexity, data scales, and model size. **Fifth**, robustness concerns encompass the sensitivity of quantized models to individual layer components, the identification of layers most susceptible to quantization, and the implications of these vulnerabilities in clinical applications (Liu et al., 2024; Hu et al., 2023).

To bridge critical knowledge gaps and address the above challenges, we present MedQuanBench, a systematic benchmark explicitly designed to evaluate quantization on CNN- and transformer-based architectures, covering representative medical imaging tasks (e.g., organ segmentation, brain segmentation), modalities (MRI, CT), and 2D/3D model variants. Specifically, we evaluate PTQ for extensive medical imaging models, demonstrating that accuracy can be substantially maintained at low precision for clinical tasks. We provide insights for clinical-relevant downstream tasks, showing how a quantized model can enable efficient real-time inference for a practical deployment scenario. The MedQuanBench explores accuracy-efficiency trade-offs, robustness to dataset/model-size and distribution shifts, and hardware performance characteristics. Our main contributions specifically address the above five critical limitations:

1. *Systematically evaluate quantization techniques on SOTA medical models:* We evaluate the efficacy of different quantization methods (e.g., smoothing, SVD, rotation) across diverse medical model architectures to identify universal and ad hoc optimization.

2. *Explore kernel and tensor compatibility:* We examine challenges in applying quantization algorithms originally designed for linear layers to 3D convolutional kernels, highlighting compatibility limitations.

3. *Analyze spatial activation variance in medical imaging:* We provide detailed analyses of spatial activation distributions and their impact on quantization accuracy, which identifies the model components that are most sensitive to spatially-driven quantization errors.

4. *Perform realistic hardware profiling:* We profile inference latency, throughput, and memory footprint across different medical imaging tasks with real hardware acceleration, offering practical insights for medical model deployment.

5. *Evaluate across scales and tasks:* We conduct the first large-scale analysis of quantization across (1) varying dataset sizes (from 100 to 10K volumes), (2) model capacities (10M to 2B parameters) (3) organ, tumor, and brain segmentation.

## 2 PRELIMINARIES

### 2.1 QUANTIZATION

**Quantization** compresses continuous floating-point values into discrete lower-bit integers, significantly reducing both computational complexity and memory requirements. This compression is crucial for medical imaging applications, where efficient model deployment is essential due to limited clinical hardware resources. In this work, we focus on symmetric uniform quantization at INT8 and INT4. Given an input floating-point tensor $X$, the quantized tensor $X_q$ is computed by:

$$X_q = \text{round}\left(\frac{X}{S}\right), \;\; S = \frac{\max(|X|)}{2^{N-1} - 1} \tag{1}$$

Here, $X_q$ denotes the integer-quantized representation of tensor $X$, and $S$ is the corresponding scaling factor computed from the tensor's maximum absolute value. For integer quantization with signed $N$-bit representations, the maximum quantized integer value is $2^{N-1} - 1$. Specifically, INT8 quantization has a maximum quantized value of 127, whereas INT4 quantization has a maximum quantized value of 7.

**Quantization granularity** refers to how many elements share a scaling factor and along which dimension(s) this sharing occurs. *Per-tensor* quantization applies a single scaling factor across the entire tensor. *Per-channel* quantization assigns individual scaling factors per output channel, effectively capturing channel-level variations. *Per-voxel* quantization assigns a unique scaling factor to each voxel, addressing spatial variations. These options are illustrated in Figure 2.

### 2.2 QUANTIZED OPERATION ON REAL HARDWARE

To be practical, quantization methods must be feasible to be mapped to supporting hardware. In this paper, we primarily target the NVIDIA Blackwell GPU, which supports 8-bit and 4-bit Microscaling (MX) data formats (Rouhani et al., 2023) in its tensor cores. At a basic level, Blackwell can efficiently perform dot products between two scaled vectors as below:

$$Y = s^{(A)} s^{(B)} (A \cdot B) \tag{2}$$

where $A$ and $B$ are 4-bit or 8-bit quantized vectors of a fixed length (32), and $S^A$ and $S^B$ are scale factors associated with each vector. Although Blackwell GPUs are not widely available yet, we briefly discuss how to efficiently map our quantization methods above to this hardware model.

For *per-tensor* quantization, the entire convolution can be done using quantized dot products and the scaling applied afterwards.

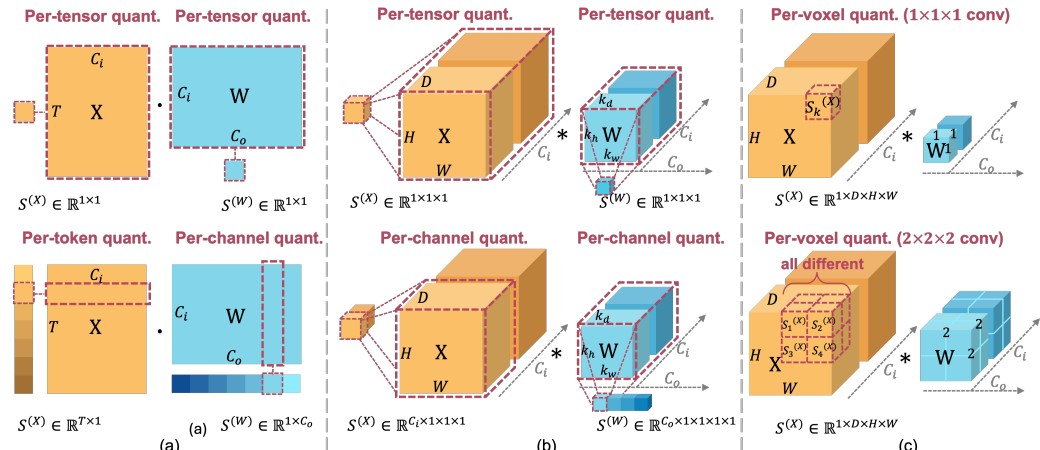

Figure 2: **Quantization granularity.** **(a) Quantization schemes for linear layers:** activation per-tensor and weight per-tensor quantization (top), activation per-token and weight per-channel quantization (bottom). Vector-wise quantization schemes (per-token, per-channel) efficiently utilize low-bit kernels when scaling factors align with outer tensor dimensions (token dimension $T$ and output channel dimension $C_o$). **(b) Quantization schemes for convolutional layers:** activation per-tensor and weight per-tensor quantization (top), activation per-channel and weight per-channel quantization (bottom). Outer tensor dimension alignment (output channel dimension $C_o$ facilitates efficient low-bit convolutional implementations. **(c) Quantization schemes for spatial dimension**: per-voxel quantization assigns unique scaling factors for each voxel. For kernel size = 1 (top), one scaling factor per voxel is sufficient; for larger kernels (bottom, shown as $2 \times 2 \times 2$), each position within the kernel uses a separate scaling factor.

For *per-channel* quantization, the convolution can be broken down into scaled dot products within each channel. For a single output voxel $Y_o$ in output channel $o$:

$$Y_o = \sum_{i=1}^{C_{in}} \left( S_i^{(X)} S_i^{(W)} \sum_{k=1}^{K} X_{i,k} W_{i,o,k} \right) \tag{3}$$

The above expresses the convolution as an outer summation over channels (indexed by $i$) and a scaled inner dot product across all spatial dims [1] (indexed by $k$ over the convolution window). $S_i^{(X)}$ and $S_i^{(W)}$ are the scale factors for channel $i$ for the input activations and the weights. Because activations are typically laid out with spatial dims last, the scaled dot product operates over contiguous data.

For *per-voxel* quantization, the convolution must be rearranged. For a single output $Y_o$ in output $o$:

$$Y_{o,k} = \sum_{k=1}^{K} \left( S_k^{(X)} S_k^{(W)} \sum_{i=1}^{C_{in}} X_{i,k} W_{i,o,k} \right) \tag{4}$$

Now the outer summation is over spatial dims in the conv window, and the scaled dot product is across the input channels. Note that for efficiency, the activations and weights must be laid out *channels-last*, which is typical. Appendix G contains run time measurements comparing activations channels-first and channels-last.

From Equation 4, we see that *per-voxel* quantization is not practical for depthwise convolutions, whose $C_{in}$ is effectively 1. This would map to quantized dot products with length 1, which is not efficient. In this paper, we use per-channel quantization for depthwise convolutions.

---

[1]The spatial dims for 3D convolutions common in medical imaging are depth, height, and width.

# 3 MEDQUANBENCH PROTOCOL

## 3.1 DATASETS AND QUANTIZATION METHODS

**Datasets.** MedQuanBench covers four representative datasets to probe quantization under different clinical conditions. BTCV (Landman et al., 2015) offers abdominal CT for multi-organ segmentation and serves as a standard robustness target. TotalSegmentator V2 (Wasserthal et al., 2023) provides whole-body CT with 117 structures for broad anatomical coverage. AbdomenAtlas 1.1 (Li et al., 2025) scales to thousands of abdominal CT volumes for dataset-size analysis. Whole Brain (Huo et al., 2019; Yu et al., 2023) contributes T1-weighted MRI with fine-grained neuroanatomy to stress small-structure segmentation. Model specifics, data splits, preprocessing, and augmentation are documented in the Appendix A.

**Quantization Methods.** Core results compare three granularity schemes across models and bit widths: *per-tensor* (one scale per layer), *per channel/token* (convolutions per-channel, linears per-token, weights per-channel), and *adaptive stratification* (per-voxel on $1 \times 1 \times 1$ convolutions, per-channel elsewhere). To examine method gains beyond granularity, activation smoothing (Xiao et al., 2023), SVD-based quantization (Li et al., 2024a), and rotation-based quantization (Ashkboos et al., 2024) are applied to the most sensitive layer identified in Sec. 4.4. Detailed configurations are provided in the Appendix B.

## 3.2 EVALUATION PROTOCOLS – ARCHITECTURES, FRAMEWORKS, METRICS

In MedQuanBench, a *backbone* denotes the high-level family (CNN or Hybrid), an *architecture* refers to the specific design and structure of a segmentation model, while a *framework* denotes a shared codebase or implementation environment supporting multiple model architectures. We evaluate representative segmentation architectures–including nnUNet (Isensee et al., 2021) and STU-Net (Huang et al., 2023b) within the nnUNet framework; UNesT (Yu et al., 2023), SwinUNETR (Tang et al., 2022), and UNETR (Hatamizadeh et al., 2022) implemented via the MONAI framework; and MedFormer (Gao et al., 2022). Segmentation performance is assessed through two widely used evaluation metrics: the Dice Similarity Coefficient (DSC), which measures overall segmentation accuracy, and the Normalized Surface Distance (NSD), which evaluates boundary alignment precision. This evaluation protocol provides consistent comparisons across models and frameworks.

## 3.3 SCALING PROTOCOLS – MODEL AND DATASET SIZE

In this study, we investigate the impact of model scale on quantization sensitivity and segmentation performance, covering a parameter range from 10M to 2B. The proposed MedQuantBench includes lightweight architectures such as nnUNet (Isensee et al., 2024) and ST-UNet-small (Huang et al., 2023b), a hybrid mid-sized model MedFormer (Gao et al., 2022), as well as multiple scales of SwinUNETR (Tang et al., 2022). This evaluation focuses on how quantization affects model capacity, particularly analyzing the trade-offs between accuracy and scaling-related factors. Overall, this work provides insights of model scale, dataset size, and robustness to low-precision weights and activations.

# 4 CORE RESULTS

We benchmark several representative 3D medical segmentation backbones pretrained on the same dataset, comparing CNN and hybrid architectures under different quantization granularities to characterize performance degradation across backbones and granularities (Sec. 4.1); we study scaling effects by evaluating the same architecture with increasing parameter counts on a fixed dataset and with varying pretraining dataset sizes, clarifying the impact of model size and dataset scale on quantization robustness (Sec. 4.2); we benchmark multiple calibration strategies and quantify their influence on post-training quantization accuracy (Sec. 4.3); we perform layer-wise analysis on both In-Distribution (ID) and Out-of-Distribution (OOD) datasets to identify quantization-sensitive layers under distribution shift, analyze INT4 failure modes on small organs, and assess advanced quantization techniques before proposing an architecture re-design guideline (Sec. 4.4); we broaden the benchmark to a lung cancer risk prediction task to increase task coverage (Sec. 4.5); and we report hardware profiling on modern GPUs to provide practical deployment insights (Sec. 4.6).

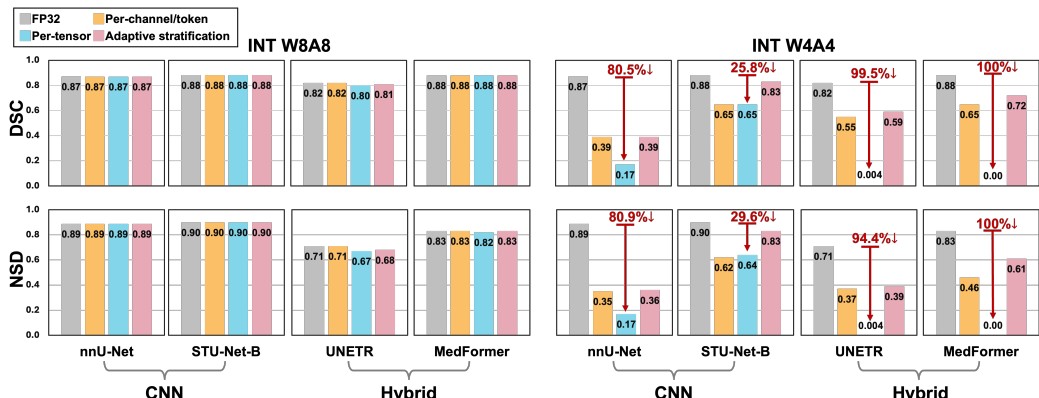

Figure 3: **Quantization results across backbones and granularities in MedQuanBench on BTCV.** INT W8A8 consistently preserves full-precision accuracy for all backbones across different granularities. In contrast, INT W4A4 performance varies with both backbone and quantization granularity: hybrid models exhibit severe collapse under per-tensor quantization, while CNNs degrade more gradually and retain information. (% ↓) indicate the relative drop from FP32.

## 4.1 BENCHMARK RESULTS ACROSS BACKBONES AND QUANTIZATION GRANULARITIES

Figure 3 summarizes quantization results across backbone and quantization granularity in MedQuan-Bench on BTCV. We compare FP32, W8A8, and W4A4 under three granularities: *per-channel/token* (conv: per-channel, linear: per-token, weights: per-channel), *per-tensor* (activations and weights share one scale per layer), and *adaptive stratification* (per-voxel for $1 \times 1 \times 1$ conv, per-channel elsewhere). INT8 remains close to FP32 for both CNN and Hybrid backbones, while INT4 depends on the different backbone and on the chosen granularity. Additional architectures see Appendix E.1

**CNNs Are Inherently More Quantization-Robust Than Hybrids:** Hybrid models such as Med-Former and UNETR are highly sensitive to low-bit quantization. Under per-tensor INT W4A4, these models show catastrophic degradation, with UNETR and MedFormer DSC dropping by 99.5% and 100%, respectively. In contrast, architectures with a CNN backbone, such as nnU-Net and STU-Net-B, degrade less severely under the same conditions, with DSC drops of 80.5% and 25.8%.. This difference likely stems from the reliance of hybrid models on linear attention and normalization layers, whose activations have high spatial variance and are difficult to capture with a single global scale.

**Finer granularity significantly improves INT W4A4 performance.** Quantization granularity has a substantial impact on segmentation accuracy. Moving from per-tensor to per-channel/token scaling consistently mitigates performance loss under INT W4A4. For hybrid models such as UNETR and MedFormer, per-tensor INT W4A4 leads to complete failure (DSC ≈ 0), whereas switching to per-channel/token granularity restores DSC to 0.55 and 0.65, respectively. Applying adaptive stratification further recover performance. These results show that finer granularity is essential for maintaining segmentation quality under low-bit settings.

## 4.2 BENCHMARK RESULTS ACROSS MODEL SCALES AND DATASET SCALES

Figure 4 summarizes MedQuanBench scaling experiments. Figure 4(a) varies model size within the SwinUNETR family on BTCV ($C = 13$, $N = 50$), where $C$ and $N$ denote the number of classes and volumes, respectively, comparing SwinUNETR-T (4.1M parameters), SwinUNETR-S (15.7M), and SwinUNETR-B (62.2M). Figure 4(b) varies dataset scale for fixed architectures: SwinUNETR-B is evaluated on BTCV ($C = 13$, $N = 50$) versus AbdomenAtlas 1.1 ($C = 25$, $N = 9,262$), and UNesT is evaluated on BTCV versus WholeBrain ($C = 133$, $N = 4,859$).

**Model size does not remedy coarse-grained INT W4A4 collapse.** In Figure 4(a), SwinUNETR-T, SwinUNETR-S, and SwinUNETR-B all exhibit complete failure under INT W4A4 per-tensor quantization, with DSC drops close to 100% regardless of model size, indicating that simply scaling capacity cannot fix the catastrophic degradation induced by coarse granularity. When switching to

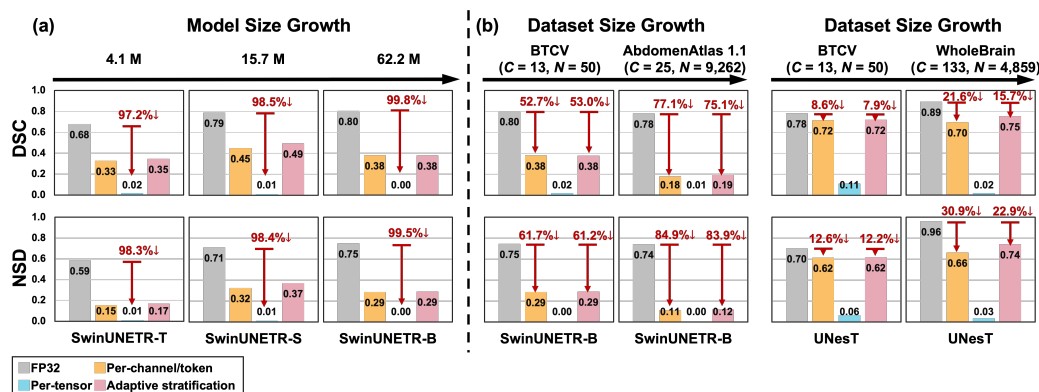

Figure 4: **4-bit quantization results across model and dataset scales in MedQuanBench.** (a) *Model scale.* We evaluate the SwinUNETR family (Tiny, Small, Base) pretrained on the same dataset to analyze how parameter count affects INT W4A4 robustness. Increasing model size does not alleviate the catastrophic degradation under coarse per-tensor quantization. (b) *Dataset scale.* Using the same architecture pretrained on datasets of increasing size and diversity, we observe larger INT W4A4 degradation.

per-channel/token or adaptive stratification, INT W4A4 performance improves substantially but does not follow a clear monotonic trend with model size.

**Larger and more diverse datasets can yield larger INT W4A4 quantization errors.** In Figure 4(b), we fix the architecture and vary the pretraining dataset. For SwinUNETR-B, moving from BTCV ($C = 13$, $N = 50$) to AbdomenAtlas 1.1 ($C = 25$, $N = 9,262$) increases the relative DSC drop under INT W4A4 per-channel/token and adaptive stratification from 52.7% and 53.0% to 77.1% and 75.1%, respectively. A similar pattern is observed for UNesT: from BTCV to Wholebrain ($C = 133$, $N = 4,859$), the relative DSC drop under INT W4A4 per-channel/token and adaptive stratification increases from 8.6% and 7.9% to 21.6% and 15.7%. These results suggest that pre-training on larger, more diverse datasets does not necessarily make models more quantization-friendly in low-bit regimes. Additional INT W8A8 results, which closely match FP32 performance across all model and dataset scales, are provided in Appendix E.2 and E.3.

## 4.3 BENCHMARK RESULTS ACROSS CALIBRATION METHODS

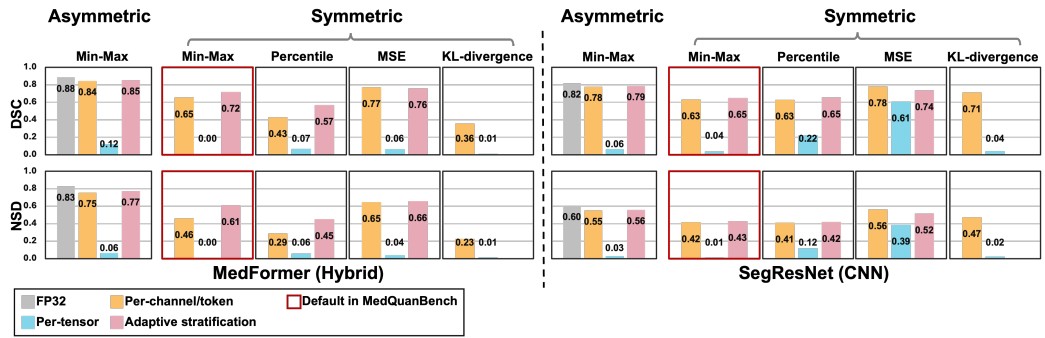

Figure 5: **4-bit quantization results across calibration methods in MedQuanBench on BTCV.** We compare INT W4A4 performance MedFormer (hybrid) and SegResNet (CNN) across multiple calibration schemes, with red boxes indicating the default MedQuanBench configuration. Adaptive stratification is not evaluated for KL-divergence due to its prohibitive calibration time.

Figure 5 evaluates INT W4A4 PTQ on BTCV for MedFormer (hybrid) and SegResNet (CNN) across multiple calibration schemes. For each model, we consider asymmetric and symmetric Min-Max, as well as symmetric percentile, MSE, and KL-based calibration under three quantization

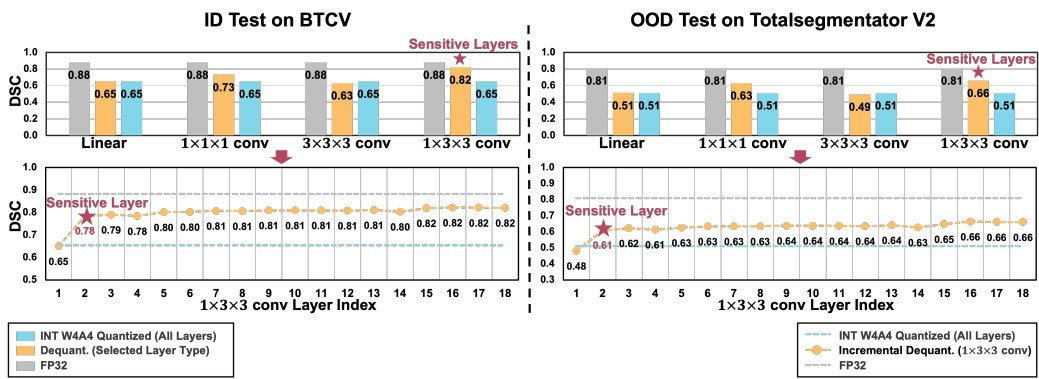

Figure 6: **Layer-wise sensitivity analysis via incremental dequantization.** We first dequantize all layers of each layer type and then incrementally dequantize individual layers within that type, performing the analysis on both ID (BTCV) and OOD (TotalSegmentator V2) data, which show consistent sensitivity patterns.

granularities. Overall, asymmetric Min-Max and symmetric MSE yield the most stable INT W4A4 performance, while percentile and especially KL-divergence exhibit larger degradation, particularly for the hybrid MedFormer. Notably, MSE calibration can substantially recover the performance of coarser granularities. For example, for SegResNet under INT W4A4 per-tensor, switching from Min-Max to symmetric MSE pushes the DSC up to 0.61. We adopt **symmetric Min-Max** as the default calibration scheme in MedQuanBench. This choice matches current 4-bit GPU formats (e.g., NVFP4/MXFP4), which use symmetric absmax scaling with shared scales per block, and avoids the zero-points and extra logic required by asymmetric quantization. We still benchmark asymmetric Min-Max and observe that it can improve INT W4A4 accuracy, but it is less naturally supported by existing 4-bit GPU formats and thus treated as an ablation rather than the deployment default.

**Calibration cost further constrains which schemes are practical.** Symmetric Min-Max and percentile can be estimated online in a single pass. For SegResNet with INT W4A4 and adaptive stratification, MSE offline calibration for one sample takes 60.9 minutes, while a full inference takes 1,114 ms. KL calibration is even more expensive: with per-voxel scales for $1 \times 1 \times 1$ convs, it requires 31.67 hours per sample. These costs make MSE and KL unsuitable as practical defaults.

### 4.4 LAYER-WISE QUANTIZATION SENSITIVITY

Quantization sensitivity varies widely across layers, and identifying the most vulnerable components is essential for reliable low-bit deployment. MedFormer was selected as the primary achitecture for the sensitivity analysis in MedQuanBench to identify the components most susceptible to quantization. This choice is motivated by MedFormer's hybrid design, which integrates convolutional kernels of sizes $1 \times 3 \times 3$ and $1 \times 1 \times 1$, along with transformer blocks employing bi-directional attention enhanced by depth-wise convolutional kernels. Its strong out-of-distribution performance on the JHH dataset reported by the Touchstone benchmark (Bassi et al., 2024) further supports its representativeness. Additional layer-wise sensitivity analysis is provided in Appendix F.

**Identifying Sensitive Layers.** Starting from a standard per-channel (convolution) and per-token (linear) quantization granularity, we gradually remove quantization from different layer types. As shown in Figure 6, dequantizing $1 \times 3 \times 3$ convolutions leads to the largest accuracy recovery for both ID and OOD test, indicating their high sensitivity. Further layer-by-layer analysis reveals that the second $1 \times 3 \times 3$ convolution is the primary bottleneck, with the most significant performance improvement after dequantization. This sensitivity can be explained by its structural position and functional role. This layer lies at a crucial junction between local convolutional features and global Transformer attention, making it particularly vulnerable to low-bit quantization. Its activations often show high dynamic range around organ boundaries, which low-bit precision struggles to represent accurately. The anisotropic $1 \times 3 \times 3$ kernel also lacks depth context, allowing quantization noise to persist. Because its outputs directly feed the attention blocks, even small errors can distort attention weights and propagate through the network, degrading segmentation quality.

Table 1: **Per-organ breakdown of INT W4A4 failures on ID (BTCV) and OOD (TotalSegmentator V2).** We report DSC and NSD for FP32, fully INT W4A4 quantized models (all layers), and incremental dequantization of the sensitive layer back to FP32.

| Organs | ID BTCV DSC FP32 | INT W4A4 | Incr. | NSD FP32 | INT W4A4 | Incr. | OOD TotalSegmentator V2 DSC FP32 | INT W4A4 | Incr. | NSD FP32 | INT W4A4 | Incr. |
|---|---|---|---|---|---|---|---|---|---|---|---|---|
| Spleen | 0.970 | 0.884 | 0.951 | 0.918 | 0.653 | 0.826 | 0.922 | 0.757 | 0.841 | 0.822 | 0.499 | 0.596 |
| Kidney R | 0.919 | 0.885 | 0.895 | 0.868 | 0.733 | 0.788 | 0.868 | 0.770 | 0.811 | 0.831 | 0.585 | 0.650 |
| Kidney L | 0.956 | 0.912 | 0.933 | 0.908 | 0.728 | 0.800 | 0.876 | 0.710 | 0.769 | 0.827 | 0.487 | 0.531 |
| Gallbladder | 0.803 | 0.100 | 0.588 | 0.732 | 0.086 | 0.344 | 0.712 | 0.099 | 0.426 | 0.626 | 0.076 | 0.176 |
| Esophagus | 0.819 | 0.675 | 0.778 | 0.770 | 0.550 | 0.695 | 0.642 | 0.427 | 0.502 | 0.585 | 0.334 | 0.410 |
| Liver | 0.976 | 0.801 | 0.878 | 0.856 | 0.323 | 0.437 | 0.939 | 0.715 | 0.777 | 0.792 | 0.275 | 0.332 |
| Stomach | 0.930 | 0.641 | 0.754 | 0.758 | 0.177 | 0.283 | 0.908 | 0.532 | 0.603 | 0.740 | 0.130 | 0.171 |
| Aorta | 0.925 | 0.859 | 0.907 | 0.870 | 0.736 | 0.820 | 0.819 | 0.624 | 0.717 | 0.761 | 0.497 | 0.567 |
| IVC | 0.879 | 0.564 | 0.653 | 0.786 | 0.206 | 0.294 | 0.836 | 0.427 | 0.437 | 0.730 | 0.131 | 0.145 |
| Portal & Splenic Vein | 0.839 | 0.488 | 0.749 | 0.810 | 0.438 | 0.633 | 0.673 | 0.302 | 0.474 | 0.663 | 0.242 | 0.355 |
| Pancreas | 0.857 | 0.614 | 0.771 | 0.745 | 0.375 | 0.540 | 0.814 | 0.491 | 0.552 | 0.718 | 0.226 | 0.287 |
| Adrenal Gland R | 0.791 | 0.524 | 0.656 | 0.860 | 0.476 | 0.623 | 0.787 | 0.401 | 0.512 | 0.868 | 0.344 | 0.434 |
| Adrenal Gland L | 0.794 | 0.549 | 0.666 | 0.857 | 0.525 | 0.686 | 0.727 | 0.372 | 0.475 | 0.784 | 0.314 | 0.431 |
| Avg | 0.882 | 0.654 | 0.783 | 0.826 | 0.462 | 0.598 | 0.809 | 0.510 | 0.607 | 0.750 | 0.318 | 0.391 |

**INT W4A4 disproportionately harms small and thin abdominal structures.** From Table 1, small-volume or thin structures such as the gallbladder, esophagus, adrenal glands, portal & splenic vein, and IVC suffer the most severe INT W4A4 degradation on both ID (BTCV) and OOD (TotalSegmentator V2). For example, gallbladder DSC drops from 0.803 to 0.100 on BTCV and from 0.712 to 0.099 on TotalSegmentator, with NSD collapsing toward zero in both cases. In contrast, large parenchymal organs such as the spleen, liver, and kidneys show much smaller relative drops. Incremental dequantization of the sensitive layer substantially restores performance for these small or thin structures (e.g., gallbladder DSC rises to 0.588 on BTCV and 0.426 on TotalSegmentator), although a clear gap to FP32 remains, especially under OOD shift.

**Limited Effect of Existing Methods.** Applying advanced PTQ methods such as activation smoothing, SVD-based, and rotation quantization yields only marginal improvements, as summarized in Table 2. This limited gain reflects a mismatch between medical activation characteristics and the assumptions of existing methods, which are built around LLM-style activation patterns where outliers cluster in a few channels and remain stable across tokens. (See activation after smoothing in Appendix B).

Table 2: **Advanced PTQ methods on the sensitive layer in MedFormer.** Starting from an INT W4A4 MedFormer with a highly sensitive $1 \times 3 \times 3$ convolution, we compare advanced PTQ methods (activation smoothing, SVD, rotation) applied to this layer, along with mixed-precision and selective dequantization. An architecture re-design that replaces the sensitive $1 \times 3 \times 3$ with a $1 \times 1 \times 1$ convolution preserves FP32 accuracy and yields the strongest INT W4A4 improvement.

| Method | Precision | Quant-Granularity | DSC | NSD |
|---|---|---|---|---|
| FP32 Baseline | FP32 | – | 0.882 | 0.826 |
| INT4 W4A4 Quantized (All layers) | INT W4A4 | per-channel/token | 0.654 | 0.462 |
| Activation Smoothing (Global) | INT W4A4 | per-channel/token | 0.650 | 0.455 |
| Activation Smoothing (Sensitive Layer) | INT W4A4 | per-channel/token | 0.648 | 0.448 |
| Activation Smoothing + SVD (Sensitive Layer) | INT W4A4 | per-channel/token | 0.569 | 0.358 |
| Rotation (Sensitive Layer) | INT W4A4 | per-channel/token | 0.621 | 0.395 |
| Mixed Precision (Sensitive Layer INT8, others INT4) | Mixed | per-channel/token | 0.661 | 0.475 |
| Selective Dequant. (Sensitive Layer FP32, others INT4) | Mixed | per-channel/token | 0.673 | 0.485 |
| | FP32 | – | 0.881 | 0.820 |
| Architecture Re-design | INT W4A4 | per-channel/token | 0.721 | 0.579 |
| | INT W4A4 | Adaptive stratification | 0.847 | 0.732 |

**Architecture-Level Optimization.** To address the identified bottleneck, we replace the most sensitive $1 \times 3 \times 3$ convolution with a $1 \times 1 \times 1$ layer, which allows finer-grained quantization. This lightweight modification preserves FP32 accuracy and improves INT4 performance from 0.654 to 0.721 under standard granularity, and up to 0.847 with adaptive stratification, approaching full-precision accuracy while remaining compatible with deployment pipelines. In addition, the $1 \times 1 \times 1$ design can be directly mapped to efficient GEMM kernels, whereas optimized 4-bit 3D conv kernels are still limited.

## 4.5 BENCHMARK RESULTS ON RISK PREDICTION TASK

We evaluate the Sybil lung cancer risk model (Mikhael et al., 2023) on the National Lung Screening Trial (NLST) dataset (NLST, 2011) for one-year lung cancer risk prediction under FP32, INT W8A8, and INT W4A4 with different granularities (Figure 7). INT W8A8 closely matches FP32 across all granularities, whereas INT W4A4 with coarse per-tensor scaling collapses to random performance, consistent with our findings.

## 4.6 HARDWARE PROFILING

While MedQuanBench primarily focuses on benchmarking low-bit quantization performance, it also provides practical deployment insights by profiling representative models on modern GPUs. Table 3 summarizes real INT8 deployment results on NVIDIA Ada architecture using TensorRT. Across different datasets and architectures, INT8 quantization consistently reduces model size by roughly $3.2{\sim}3.8\times$, accelerates inference by about $2.1{\sim}2.7\times$ and reduces GPU memory by about $1.04{\sim}1.74\times$ while maintaining segmentation performance nearly identical to FP32. These results confirm that 8-bit quantization is a stable and deployment-ready solution for medical imaging

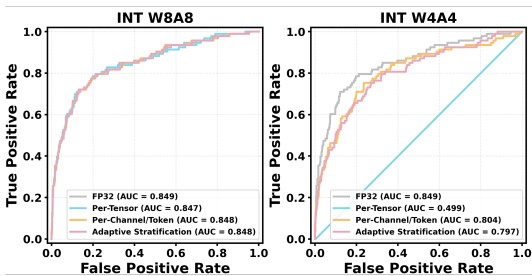

Figure 7: **ROC curves for lung cancer risk prediction under INT W8A8 and W4A4.**

models in clinical settings. As medical segmentation models and datasets continue to grow in size and complexity, reducing model size and latency becomes increasingly important for mitigating memory and throughput bottlenecks in clinical deployment.

Table 3: **Quantization results on modern GPUs.** INT8 deployment performance of representative medical segmentation models on NVIDIA Ada GPUs using TensorRT. Compared with FP32, INT8 consistently reduces model size by up to $3.8\times$, accelerates inference by up to $2.7\times$, and reduces GPU memory by up to $1.7\times$, while maintaining accuracy, demonstrating its readiness for clinical deployment. As model and dataset scales increase, such compression is crucial for practical applications. Emerging platforms such as NVIDIA Blackwell, which provide native sub-8-bit support (e.g., 4 bit), enable efficiency gains.

| Dataset | Architecture | Model Size (MB) | | Latency (ms) | | GPU Memory (MB) | |
|---|---|---|---|---|---|---|---|
| | | FP32 | INT W8A8 (Reduction Ratio) | FP32 | INT W8A8 (Latency Gain) | FP32 | INT W8A8 (Mem. Reduction) |
| BTCV | U-Net (2015) | 23.11 | 6.61 (3.50×) | 2.62 | 1.05 (2.50×) | 792 | 686 (1.15×) |
| | TransUNet (2021) | 351.85 | 91.90 (3.83×) | 4.09 | 1.74 (2.35×) | 1,134 | 772 (1.47×) |
| WholeBrain | UNesT (2023) | 349.41 | 96.72 (3.61×) | 5.59 | 2.72 (2.06×) | 1,230 | 938 (1.31×) |
| TotalSeg V2 | STU-Net-S (2023b) | 55.7 | 20.5 (2.72×) | 2.6 | 1.0 (2.60×) | 1,820 | 1,742 (1.04×) |
| | STU-Net-H (2023b) | 5,559.4 | 1,519.8 (3.66×) | 98.5 | 30.2 (3.26×) | 9,384 | 5,394 (1.74×) |
| | nnU-Net (2021) | 107.84 | 33.97 (3.17×) | 2.99 | 1.25 (2.39×) | 1,085 | 760 (1.43×) |
| | SwinUNETR (2021) | 247.96 | 70.18 (3.53×) | 9.85 | 3.59 (2.74×) | 2,374 | 2,114 (1.12×) |
| | SegResNet (2019) | 170.44 | 50.29 (3.39×) | 5.14 | 2.06 (2.49×) | 1,796 | 1,460 (1.23×) |
| | VISTA3D (2024) | 264.57 | 71.18 (3.72×) | 4.59 | 1.93 (2.38×) | 2,558 | 1,852 (1.38×) |

## 5 CONCLUSION

Quantization presents a promising path for improving the deployment of medical AI models in resource-constrained clinical environments, such as edge GPUs, hospital CPUs, and remote healthcare systems. By reducing memory footprint and enhancing computational efficiency, quantized models facilitate time-sensitive medical tasks. MedQuanBench reveals that while 8-bit quantization is generally robust and 4-bit precision demands careful granularity control to preserve accuracy. Our sensitivity analysis further identifies architectural components most vulnerable to quantization, providing actionable insights for balancing precision, efficiency, and reliability in deployment.

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

# Appendix

## A    DATASET DESCRIPTIONS

MedQuanBench incorporates four carefully selected datasets, consisting of diverse imaging modalities, anatomical regions, and annotation granularities, to evaluate quantization techniques across realistic medical scenarios.

**AbdomenAtlas 1.1**(Li et al., 2025) comprises 9,262 abdominal CT scans collected globally from 238 hospitals, annotated at voxel-level for 25 abdominal organs. It is used as a dataset scaling analysis resource, ensuring quantization technology efficacy on diverse clinical settings. (see Table 4)

**BTCV (Beyond the Cranial Vault)** (Landman et al., 2015; Tang et al., 2021) includes 50 abdominal CT volumes annotated for 13 key organ and vessel structures. 30 scans are exploited for model training and validation, while the remaining 20 scans serve as testing cases in MedQuanBench. The dataset originates from clinical research studies at Vanderbilt University Medical Center on healthy anatomies, providing a high-quality dataset to test model performance on a well-defined segmentation task.

**TotalSegmentator V2** (Wasserthal et al., 2022) provides extensive anatomical coverage with 1,228 full-body CT scans annotated for 117 anatomical structures (brain, organs, bones, vessels). Scans originate from multiple institutes within the University Hospital Basel network. MedQuanBench utilizes a distinct subset ( 743 scans) exclusively for evaluation, representing a rigorous test of model robustness and generalization to unseen clinical populations and imaging conditions.

**Whole Brain Segmentation Dataset** (Huo et al., 2019; Yu et al., 2023) consists of MRI T1-weighted volumes acquired from multiple institutions, structured specifically for detailed neuroanatomical segmentation. It includes a primary manually annotated training set (50 MRI scans from the OASIS dataset, labeled with 133 brain regions) and two distinct evaluation sets: the high-resolution Colin27 scan (labeled with 130 regions) and 13 pediatric scans from the CANDI dataset (ages 5-15, labeled with 130 regions). Additionally, MedQuanBench incorporates an auxiliary dataset of 4,859 MRI scans automatically segmented using multi-atlas techniques for large-scale pretraining before fine-tuning with manually labeled OASIS data. This design enables assessment across age groups, resolutions, and labeling granularities, testing quantization robustness in fine-grained neuro-imaging tasks. (see Table 5)

Table 4: **Public Datasets Comprising AbdomenAtlas 1.1.** Constructed from 17 publicly available datasets (items 1-17), it comprises 9,262 abdominal CT volumes with 25 annotated classes per volume. Due to overlapping volumes among sources, the total count does not equal the sum of individual datasets. Its diversity–spanning 88 centers across 9 countries–makes it ideal for evaluating quantization robustness in varied clinical settings.

| Dataset | # of classes | # of volumes | # of centers | source countries | license |
|---|---|---|---|---|---|
| 1. Pancreas-CT (2015) | 1 | 42 | 1 | US | CC BY 3.0 |
| 2. LiTS (2019) | 1 | 131 | 7 | DE, NL, CA, FR, IL | CC BY-SA 4.0 |
| 3. KiTS (2020) | 1 | 489 | 1 | US | CC BY-NC-SA 4.0 |
| 4. AbdomenCT-1K (2021) | 4 | 1,050 | 12 | DE, NL, CA, FR, IL, US, CN | CC BY-NC-SA |
| 5. CT-ORG (2020) | 5 | 140 | 8 | DE, NL, CA, FR, IL, US | CC BY 3.0 |
| 6. CHAOS (2018) | 4 | 20 | 1 | TR | CC BY-SA 4.0 |
| 7-12. MSD CT Tasks (2021) | 9 | 945 | 1 | US | CC BY-SA 4.0 |
| 13. BTCV (2015) | 12 | 50 | 1 | US | CC BY 4.0 |
| 14. AMOS22 (2022) | 15 | 200 | 2 | CN | CC BY-NC-SA |
| 15. WORD (2021) | 16 | 120 | 1 | CN | GNU GPL 3.0 |
| 16. FLARE'23 | 13 | 4,100 | 30 | - | CC BY-NC-ND 4.0 |
| 17. Abdominal Trauma Det (2023) | 0 | 4714 | 23 | - | - |
| 18. AbdomenAtlas 1.1 (2025) | 25 | 9,262 | 88 | US, DE, NL, FR, IL, CN, CA, TR, CH | - |

US: United States    DE: Germany    NL: Netherlands    CA: Canada    FR: France    IL: Israel
CN: China    TR: Turkey    CH: Switzerland

Table 5: **Public Neuroimaging Datasets Comprising WholeBrain.** WholeBrain aggregates 4,859 brain MRI volumes from eight publicly available, multi-center datasets. By capturing diverse neuroanatomical segmentation scenarios, it complements abdominal CT benchmarks and strengthens quantization evaluation across distinct clinical modalities.

| Study Name | Website | # of Volumes |
|---|---|---|
| Attention Deficit Hyperactivity Disorder (ADHD200) | fcon_1000.projects.nitrc.org/indi/adhd200 | 950 |
| Autism Brain Imaging Data Exchange (ABIDE) | fcon_1000.projects.nitrc.org/indi/abide | 563 |
| Baltimore Longitudinal Study of Aging (BLSA) | www.blsa.nih.gov | 614 |
| Cutting Pediatrics | vkc.mc.vanderbilt.edu/ebrl | 586 |
| Information Extraction from Images (IXI) | www.nitrc.org/projects/ixi_dataset | 541 |
| Nathan Kline Institute Rockland (NKI_rockland) | fcon_1000.projects.nitrc.org/indi/enhanced | 141 |
| Open Access Series of Imaging Studies (OASIS) | www.oasis-brains.org | 312 |
| 1000 Functional Connectome (fcon_1000) | fcon_1000.projects.nitrc.org | 1102 |
| **WholeBrain (Total)** | — | **4859** |

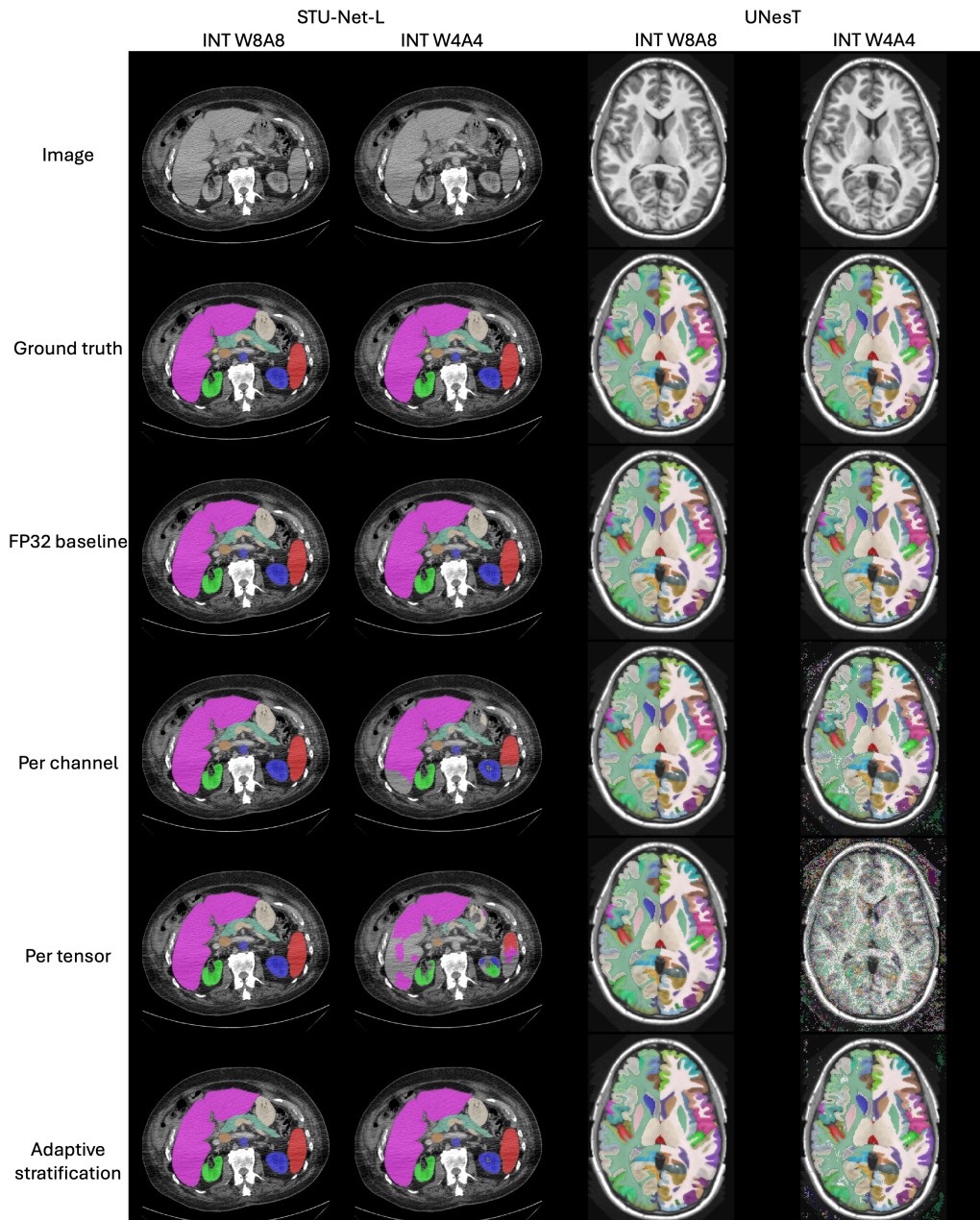

Figure 8: **Visual Comparison of Quantization Results on BTCV and WholeBrain Datasets.** *Left two columns:* STU-Net-L segmentation results on BTCV dataset at different precision levels (INT W8A8 and INT W4A4) and quantization granularities (per-channel, per-tensor, adaptive stratification). *Right two columns:* UNesT segmentation predictions on WholeBrain dataset under the same quantization settings. 8-bit quantization results closely align with the FP32 baseline, demonstrating minimal accuracy loss. However, 4-bit quantization shows a notable variation in performance, with higher quantization granularity (e.g., adaptive stratification) yielding better segmentation quality compared to lower granularity methods (e.g., per-tensor).

## B QUANTIZATION METHODS

MedQuanBench evaluates three representative quantization methods–smoothing, SVD-based decomposition, and rotation–each targeting distinct quantization challenges through different approaches.

**Smoothing** (Xiao et al., 2023) addresses the challenge of activation outliers, which can hinder quantization by distorting numeric ranges. This method redistributes activation magnitudes between activations and weights using complementary scaling factors. Specifically, extreme values are scaled downward, while corresponding weights are scaled upward, preserving the original model computation. By balancing activation distributions, smoothing reduces quantization errors caused by outliers.

**SVD-based Low-Rank Decomposition** (Li et al., 2024a) targets outlier values within weight matrices. The method factorizes weights into low-rank approximations, separating significant outlier components from the rest. A small set of high-magnitude components is retained at higher precision or handled separately, while the remaining weights are quantized directly. This decomposition isolates problematic weight values, making the overall weight quantization more uniform and less error-prone.

**Rotation-based Transform** (Ashkboos et al., 2024) focuses on balancing uneven value distributions in activations or weights. It applies an orthogonal transformation (rotation) to redistribute values across multiple dimensions. The rotated representation facilitates efficient low-bit quantization by spreading large outliers more evenly. After quantization, an inverse rotation restores the original computational form, ensuring mathematical equivalence to the original model computation.

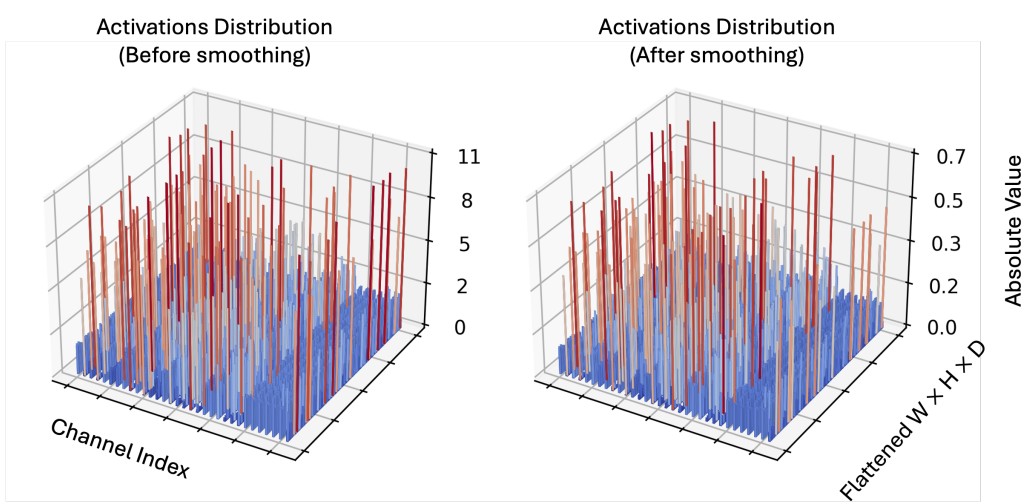

Figure 9: **Activation distribution before and after smoothing on a representative medical segmentation model layer.** The two subplots visualize the absolute value of input activations to a quantization-sensitive layer, arranged by channel (x-axis) and spatial position (y-axis). The left plot shows the distribution before smoothing, characterized by sharp outliers within many channels. The right plot shows the result after smoothing using $\alpha = 0.5$, where activation magnitudes are reduced. While prior works show that outliers persistently dominate specific channels (Xiao et al., 2023), medical models display a different pattern: outliers are unevenly distributed *within* each channel, rather than fixed across all spatial positions or tokens. This structural discrepancy suggests that channel-wise smoothing alone is insufficient for handling activation outliers in medical models. Instead, outliers frequently manifest across channels at specific spatial sites, limiting the effectiveness of conventional smoothing and highlighting the need for finer-grained or cross-channel quantization strategies.

## C   KURTOSIS AND OUTLIERS

Kurtosis ($\kappa$) is the standardized fourth central moment of a distribution, defined mathematically as

$$\kappa = \mathbb{E}\left[\left(\frac{X - \mu}{\sigma}\right)^4\right], \tag{5}$$

where $X$ is a random variable, $\mu$ is its mean, and $\sigma$ its standard deviation. This metric quantifies the *tailedness* of a distribution, indicating how heavy or light the tails are compared to a normal distribution (which has $\kappa = 3$). High kurtosis values show the presence of extreme outliers within the distribution. Empirical observations in medical imaging models show greater kurtosis within channels than across channels. This suggests that extreme activation outliers occur within individual channels rather than uniformly across channels. This result highlights potential limitations of channel-wise normalization or per-channel quantization strategies for medical models. On the contrary, across-channel kurtosis tends to be lower, which indicates more stable distributions across the channel dimension at each spatial location. This observation motivates the use of per-voxel quantization, which assigns a single scaling factor to all channels at each spatial position. Thus, it can better align with the observed activation distributions. However, this approach can introduce large computational overhead due to the large number of required scaling factors, especially given that the spatial dimensions in medical imaging models are typically larger than the number of channels. As a result, the choice between per-channel and per-voxel quantization strategies involves a fundamental trade-off between preserving accuracy and maintaining computational efficiency.

## D   QUANTIZATION: RELATED WORKS

Model quantization is an emerging technique for accelerating and deploying AI models on certain hardware, particularly in LLM, computer vision, and recently, medical imaging domains. Quantization methods are broadly categorized into Quantization-Aware Training (QAT) and Post-Training Quantization (PTQ). QAT includes a low-precision or mix-precision simulation during training, which enables models to align quantization-induced bias via methods like Straight-Through Estimators (STE) (Yin et al., 2019) or differentiable quantization algorithm (Zhou et al., 2016). MedQ (Zhang & Chung, 2021) proposes an ultra-low-bit QAT framework for U-Net, achieving ternary (2-bit) quantization on 3D medical segmentation tasks and (AskariHemmat et al., 2019) present a fixed-point QAT method for 2D U-Net, demonstrating theoretical memory reduction with 4-bit weights and 6-bit activations. While effective, QAT demands access to full training datasets, which is often limited and unexplored in medical imaging due to data challenges (Price & Cohen, 2019) and the scale of datasets like huge volumes set AbdomenAtlas (Qu et al., 2023). In addition, PTQ, requires no retraining or uses minimal unlabeled calibration data to adjust pre-trained models, making it more capable for clinical practice. Recent works like AdaRound (Nagel et al., 2020) and BRECQ (Li et al., 2021) have PTQ for certain layers by optimizing weight rounding and layer-wise dependencies, while methods such as PTQ4ViT (Yuan et al., 2022) can address challenges in quantizing vision transformers (ViTs), such as post-softmax distribution skew and activation outliers. EfficientQ (Zhang & Chung, 2024) introduces an ADMM-based PTQ algorithm for medical segmentation, focusing on efficient calibration with minimal data samples. However, existing PTQ approaches are only fake quantization, which simulates low-precision computation during inference but maintains high-precision weights and activations in memory, yielding no real reductions in model size or latency (Gholami et al., 2022). Recent study(Qu et al., 2025) provided INT8 real deployment using TensorRT.

In medical imaging, where 3D segmentation models such as U-Net (Ronneberger et al., 2015), Swin-UNETR (Hatamizadeh et al., 2021), and STU-Net (Huang et al., 2023b) demand high computational resources, the gap between simulated and real quantization efficiency becomes critical. Prior efforts have been made to balance accuracy preservation with actual deployment gains. For instance, fake quantization of ViTs in PTQ4SAM (Lv et al., 2024) improved attention map quantization but failed to reduce memory footprint. This limitation is even intense by the growing scale of medical datasets (e.g., TotalSegmentator V2 (Wasserthal et al., 2023) with 117 labels) and models, where large-scale architectures like VISTA3D (He et al., 2024) require efficient inference speed and memory footprint. Recent frameworks like TensorRT offer promise by enabling hardware-accelerated real quantization, converting models to INT8 precision with true memory and latency savings. However, systematic exploration of real PTQ applicability to diverse medical segmentation architectures remains limited,

which leaves a critical need for frameworks that bridge the divide between theoretical quantization benefits and clinical utility.

In this work, we explore three representative quantization techniques–activation smoothing (Xiao et al., 2023), singular value decomposition (SVD)-based quantization (Li et al., 2024a), and rotation quantization (Ashkboos et al., 2024)–as initial attempts to quantify their effectiveness on medical models, particularly focusing on layers identified as sensitive to quantization-induced errors. Specifically, we adopt an activation smoothing factor of $\alpha = 0.5$ to balance the redistribution of extreme activations between activations and weights. For SVD-based quantization, we utilize a low-rank approximation with rank set to 4, isolating significant weight outliers to enhance quantization stability. Additionally, rotation quantization is implemented via a Hadamard matrix of order 32, matching the input channel dimension of the quantization-sensitive layer.

# E  ADDITIONAL ANALYSIS OF BENCHMARK EXPERIMENTS

## E.1  QUANTIZATION RESULTS OF DIFFERENT BACKBONES

Table 6: **Quantization results across backbones and granularities in MedQuanBench on BTCV** FP32, INT8, and INT4 evaluated under per-tensor, per-channel/token, and adaptive stratification INT8 is close to FP32 across models. INT4 varies with backbone and granularity, CNNs degrade more gradually than Hybrids, and finer granularity improves robustness. Cells report DSC/NSD with relative drop ($\downarrow\Delta$ %) vs FP32.

| Backbone | Architectures | Precision | Quant-Granularity | DSC ($\downarrow\Delta$ %) | NSD ($\downarrow\Delta$ %) |
|---|---|---|---|---|---|
| CNN | | FP32 | – | 0.872 (–) | 0.888 (–) |
| | nnU-Net (2021) | INT W8A8 | Per-channel | 0.870 (0.2%) | 0.887 (0.1%) |
| | | | Per-tensor | 0.870 (0.2%) | 0.888 (0) |
| | | | Adaptive stratification | 0.870 (0.2%) | 0.887 (0.1%) |
| | | INT W4A4 | Per-channel | 0.387 (55.6%) | 0.354 (60.1%) |
| | | | Per-tensor | 0.170 (80.5%) | 0.169 (80.9%) |
| | | | Adaptive stratification | 0.393 (54.9%) | 0.358 (59.7%) |
| | STU-Net-B (2023b) | FP32 | – | 0.881 (–) | 0.903 (–) |
| | | INT W8A8 | Per-channel | 0.881 (0) | 0.901 (0.2%) |
| | | | Per-tensor | 0.881 (0) | 0.902 (0.1%) |
| | | | Adaptive stratification | 0.881 (0) | 0.902 (0.1%) |
| | | INT W4A4 | Per-channel | 0.647 (26.6%) | 0.619 (31.5%) |
| | | | Per-tensor | 0.654 (25.8%) | 0.636 (29.6%) |
| | | | Adaptive stratification | 0.829 (5.9%) | 0.833 (7.8%) |
| Hybrid | SwinUNETR (2022) | FP32 | – | 0.849 (–) | 0.760 (–) |
| | | INT W8A8 | Per-channel/token | 0.849 (0) | 0.761 (1% ↑) |
| | | | Per-tensor | 0.849 (0) | 0.761 (1% ↑) |
| | | | Adaptive stratification | 0.849 (0) | 0.761 (1% ↑) |
| | | INT W4A4 | Per-channel/token | 0.565 (33.5%) | 0.446 (41.3%) |
| | | | Per-tensor | 0.059 (93.1%) | 0.054 (92.9%) |
| | | | Adaptive stratification | 0.571 (32.7%) | 0.447 (41.2%) |
| | UNETR (2022) | FP32 | – | 0.824 (–) | 0.714 (–) |
| | | INT W8A8 | Per-channel/token | 0.824 (0) | 0.714 (0) |
| | | | Per-tensor | 0.802 (2.7%) | 0.669 (6.3%) |
| | | | Adaptive stratification | 0.809 (1.8%) | 0.676 (5.3%) |
| | | INT W4A4 | Per-channel/token | 0.553 (35.3%) | 0.366 (48.7%) |
| | | | Per-tensor | 0.004 (99.5%) | 0.004 (94.4%) |
| | | | Adaptive stratification | 0.590 (28.4%) | 0.386 (45.9%) |
| | MedFormer (2022) | FP32 | – | 0.882 (–) | 0.826 (–) |
| | | INT W8A8 | Per-channel/token | 0.882 (0) | 0.826 (0) |
| | | | Per-tensor | 0.880 (0.2%) | 0.823 (0.3%) |
| | | | Adaptive stratification | 0.882 (0) | 0.826 (0) |
| | | INT W4A4 | Per-channel/token | 0.654 (25.9%) | 0.462 (44.1%) |
| | | | Per-tensor | 0.000 (100%) | 0.000 (100%) |
| | | | Adaptive stratification | 0.719 (18.5%) | 0.610 (26.3%) |
| | MedSam2 (2025) | FP32 | – | 0.928 (–) | 0.886 (–) |
| | | INT W8A8 | Per-channel/token | 0.926 (0.2%) | 0.877 (1.0%) |
| | | | Adaptive stratification | 0.924 (0.4%) | 0.873 (1.5%) |
| | | | Per-tensor | 0.921 (0.8%) | 0.867 (2.1%) |
| | | INT W4A4 | Per-channel/token | 0.011 (98.8%) | 0.003 (99.7%) |
| | | | Adaptive stratification | 0.010 (98.9%) | 0.003 (99.7%) |
| | | | Per-tensor | 0.026 (97.2%) | 0.067 (92.4%) |

## E.2 Quantization Results under Model Scaling

Table 7: **Quantization results across model scales on BTCV.** We evaluate STU-Net (Base/Large/Huge) and SwinUNETR (Tiny/Small/Base) models with increasing parameter sizes to assess whether model scale influences quantization robustness. Across all models, INT8 quantization maintains segmentation performance nearly identical to the FP32 baseline. However, the sensitivity to INT4 quantization does not show a consistent trend with model size: larger models are not strictly more or less robust. Instead, quantization granularity emerges as a more reliable factor, as adaptive stratification consistently improves performance over lower-granularity schemes, highlighting its importance in achieving accurate low-bit deployment in medical imaging.

| Framework | Architecture | Backbone | Param | Precision | Quant-Granularity | DSC ($\downarrow\Delta$%) | NSD ($\downarrow\Delta$%) |
|---|---|---|---|---|---|---|---|
| nnUNet | STU-Net-B (2023b) | CNN | 58.3 M | FP32 | – | 0.881 (–) | 0.903 (–) |
| | | | | INT W8A8 | Per-channel | 0.881 (0) | 0.901 (0.2%)) |
| | | | | | Per-tensor | 0.881 (0) | 0.902 (0.1%) |
| | | | | | Adaptive stratification | 0.881 (0) | 0.902 (0.1%) |
| | | | | INT W4A4 | Per-channel | 0.647 (26.6%) | 0.619 (31.5%) |
| | | | | | Per-tensor | 0.654 (25.8%) | 0.636 (29.6%) |
| | | | | | Adaptive stratification | 0.829 (5.9%) | 0.833 (7.8%) |
| | STU-Net-L (2023b) | CNN | 440.3 M | FP32 | – | 0.880 (–) | 0.903 (–) |
| | | | | INT W8A8 | Per-channel | 0.880 (0) | 0.902 (0.1%) |
| | | | | | Per-tensor | 0.880 (0) | 0.903 (0) |
| | | | | | Adaptive stratification | 0.880 (0) | 0.902 (0.1%) |
| | | | | INT W4A4 | Per-channel | 0.701 (20.3%) | 0.695 (23.0%) |
| | | | | | Per-tensor | 0.466 (47.0%) | 0.460 (49.1%) |
| | | | | | Adaptive stratification | 0.857 (2.6%) | 0.870 (3.7%) |
| | STU-Net-H (2023b) | CNN | 1,457.3 M | FP32 | – | 0.873 (–) | 0.889 (–) |
| | | | | INT W8A8 | Per-channel | 0.873 (0) | 0.889 (0) |
| | | | | | Per-tensor | 0.872 (0.1%) | 0.889 (0) |
| | | | | | Adaptive stratification | 0.872 (0.1%) | 0.889 (0) |
| | | | | INT W4A4 | Per-channel | 0.700 (19.8%) | 0.681 (23.4%) |
| | | | | | Per-tensor | 0.734 (15.9%) | 0.716 (19.5%) |
| | | | | | Adaptive stratification | 0.840 (3.8%) | 0.848 (4.6%) |
| MONAI | SwinUNETR-T (2022)hybrid | | 4.1 M | FP32 | – | 0.684 (–) | 0.586 (–) |
| | | | | INT W8A8 | Per-channel/token | 0.682 (0.3%) | 0.583 (0.5%)) |
| | | | | | Per-tensor | 0.679 (0.7%) | 0.578 (1.4%) |
| | | | | | Adaptive stratification | 0.683 (0.1%) | 0.584 (0.3%) |
| | | | | INT W4A4 | Per-channel/token | 0.328 (52.0%) | 0.154 (73.7%) |
| | | | | | Per-tensor | 0.019 (97.2%) | 0.010 (98.3%) |
| | | | | | Adaptive stratification | 0.347 (49.2%) | 0.169 (71.2%) |
| | SwinUNETR-S (2022)hybrid | | 15.7 M | FP32 | – | 0.788 (–) | 0.713 (–) |
| | | | | INT W8A8 | Per-channel/token | 0.787 (0.1%) | 0.712 (0.1%) |
| | | | | | Per-tensor | 0.783 (0.6%) | 0.704 (1.2%) |
| | | | | | Adaptive stratification | 0.787 (0.1%) | 0.713 (0) |
| | | | | INT W4A4 | Per-channel/token | 0.450 (42.9%) | 0.324 (54.5%) |
| | | | | | Per-tensor | 0.012 (98.5%) | 0.011 (98.4%) |
| | | | | | Adaptive stratification | 0.494 (37.3%) | 0.371 (48.0%) |
| | SwinUNETR-B (2022)hybrid | | 62.2 M | FP32 | – | 0.804 (–) | 0.746 (–) |
| | | | | INT W8A8 | Per-channel/token | 0.803 (0.1%) | 0.744 (0.3%) |
| | | | | | Per-tensor | 0.802 (0.2%) | 0.740 (0.8%) |
| | | | | | Adaptive stratification | 0.804 (0) | 0.745 (0.1%) |
| | | | | INT W4A4 | Per-channel/token | 0.380 (52.7%) | 0.286 (61.7%) |
| | | | | | Per-tensor | 0.002 (99.8%) | 0.004 (99.5%) |
| | | | | | Adaptive stratification | 0.378 (53.0%) | 0.289 (61.2%) |

## E.3 QUANTIZATION RESULTS ACROSS DIFFERENT MODALITY AND DATASETS

Table 8: **Quantization results across modalities and dataset scales.** This table evaluates the quantization robustness of UNesT and SwinUNETR across datasets varying in imaging modality, class numbers, and scale. BTCV and AbdomenAtlas 1.1 are both abdominal CT segmentation datasets, while WholeBrain involves brain MRI. The datasets also vary significantly in size and complexity: BTCV includes 50 CT volumes with 13 labeled abdominal structures, AbdomenAtlas 1.1 contains 9,262 CT volumes with 25 anatomical labels, and WholeBrain comprises 4,859 MRI volumes covering 133 fine-grained brain regions. As the dataset size and class number increase, models show greater sensitivity to 4 bit quantization. For instance, under Per-channel/token quantization granularity, UNesT shows a minor DSC drop of 8.6% on BTCV, but a more substantial 21.9% drop on WholeBrain. Similarly, SwinUNETR's DSC drops 52.7% on BTCV, compared to 77.1% on AbdomenAtlas. These findings highlight the increasing challenge of low-bit quantization under high-resolution, large-scale conditions, and underscore the importance of employing finer granularity or more adaptive quantization strategies in such settings.

| Architecture | Backbone | Param | Dataset | Precision | Quant-Granularity | DSC ($\downarrow \Delta$ %) | NSD ($\downarrow \Delta$ %) |
|---|---|---|---|---|---|---|---|
| UNesT | Hybrid | 87.3 M | BTCV | FP32 | – | 0.783 (–) | 0.704 (–) |
| | | | | INT W8A8 | Per-channel/token | 0.783 (0) | 0.704 (0) |
| | | | | | Per-tensor | 0.783 (0) | 0.702 (0.3%) |
| | | | | | Adaptive stratification | 0.783 (0) | 0.704 (0) |
| | | | | INT W4A4 | Per-channel/token | 0.716 (8.6%) | 0.615 (12.6%) |
| | | | | | Per-tensor | 0.111 (85.8%) | 0.064 (90.9%) |
| | | | | | Adaptive stratification | 0.721 (7.9%) | 0.618 (12.2%) |
| | | | WholeBrain | FP32 | – | 0.893 (–) | 0.961 (–) |
| | | | | INT W8A8 | Per-channel/token | 0.893 (0) | 0.961 (0)) |
| | | | | | Per-tensor | 0.887 (0.6%) | 0.959 (0.2%) |
| | | | | | Adaptive stratification | 0.893 (0) | 0.961 (0) |
| | | | | INT W4A4 | Per-channel/token | 0.697 (21.9%) | 0.664 (30.9%) |
| | | | | | Per-tensor | 0.019 (97.8%) | 0.034 (96.5%) |
| | | | | | Adaptive stratification | 0.753 (15.7%) | 0.741 (22.9%) |
| SwinUNETR | Hybrid | 62.2 M | AbdomenAtlas 1.1 | FP32 | – | 0.780 (–) | 0.742 (–) |
| | | | | INT W8A8 | Per-channel/token | 0.779 (0.1%) | 0.741 (0.1%) |
| | | | | | Per-tensor | 0.773 (0.9%) | 0.731 (1.5%) |
| | | | | | Adaptive stratification | 0.779 (0.1%) | 0.741 (0.1%) |
| | | | | INT W4A4 | Per-channel/token | 0.179 (77.1%) | 0.112 (84.9%) |
| | | | | | Per-tensor | 0.006 (99.2%) | 0.004 (99.5%) |
| | | | | | Adaptive stratification | 0.194 (75.1%) | 0.119 (83.9%) |
| | | | BTCV | FP32 | – | 0.804 (–) | 0.746 (–) |
| | | | | INT W8A8 | Per-channel/token | 0.803 (0.1%) | 0.744 (0.3%) |
| | | | | | Per-tensor | 0.802 (0.2%) | 0.740 (0.8%) |
| | | | | | Adaptive stratification | 0.804 (0) | 0.745 (0.1%) |
| | | | | INT W4A4 | Per-channel/token | 0.380 (52.7%) | 0.286 (61.7%) |
| | | | | | Per-tensor | 0.002 (99.8%) | 0.004 (99.5%) |
| | | | | | Adaptive stratification | 0.378 (53.0%) | 0.289 (61.2%) |

# F    ADDITIONAL LAYER-WISE QUANTIZATION SENSITIVITY

We further provide supplementary analyses on SegFormer3D (Perera et al., 2024) to validate the layer-wise sensitivity findings in Sec. 4.4. Table 9 presents an incremental dequantization experiment, while Table 10 benchmarks common PTQ methods applied specifically to the most sensitive layer.

Table 9: **Incremental dequantization analysis on SegFormer3D.** We incrementally remove INT4 quantization from individual $3 \times 3 \times 3$ convolution layers to assess their relative contribution to overall accuracy degradation. The fourth convolution layer shows the largest recovery in DSC and NSD, indicating it as the most quantization-sensitive component.

| Incremental Dequantization | DSC | NSD |
|---|---|---|
| FP32 Baseline | 0.815 | 0.782 |
| INT4 Quantized (All layers) | 0.767 | 0.708 |
| INT4 Quantized exclude 1st $3\times3\times3$ conv | 0.765 | 0.706 |
| INT4 Quantized exclude 2nd $3\times3\times3$ conv | 0.766 | 0.709 |
| INT4 Quantized exclude 3rd $3\times3\times3$ conv | 0.768 | 0.716 |
| **INT4 Quantized exclude 4th** $3\times3\times3$ **conv** | **0.778** | **0.718** |

Table 10: **Quantization performance on sensitive layers of SegFormer3D.** We apply advanced PTQ methods to the most sensitive layer identified in Table 9. Activation smoothing and SVD-based decomposition yield marginal gains, highlighting the challenge of quantizing activation distributions in medical segmentation.

| Method | Precision | Quant-Granularity | DSC | NSD |
|---|---|---|---|---|
| Baseline | FP32 | – | 0.815 | 0.782 |
| INT4 Quantized (All layers) | INT W4A4 | per-channel/token | 0.767 | 0.708 |
| Activation Smoothing | INT W4A4 | per-channel/token | 0.771 | 0.711 |
| Activation Smoothing + SVD | INT W4A4 | per-channel/token | 0.769 | 0.698 |

# G  ADDITIONAL HARDWARE PROFILING

Table 11: **Latency comparison under different memory layouts.** This table compares inference latency of the MedFormer model using channel-first (default in most deep learning frameworks for convolutional operations) and channel-last layouts, under a fixed input patch size of $32 \times 128 \times 128$. Although low-bit quantization schemes such as per-voxel scaling benefit from channel-last layouts due to more contiguous memory access across spatial dimensions, most convolutional backends (e.g., cuDNN) remain optimized for channel-first formats. However, we observe minimal latency differences between the two layouts in our setting.

| | | Latency (ms) | |
|---|---|---|---|
| Architecture | Patch size | Channel-first $[B \times C \times D \times H \times W]$ | Channel-last $[B \times D \times H \times W \times C]$ |
| Medformer | $[32 \times 128 \times 128]$ | 85.3 | 86.6 |

# H  ADDITIONAL TASK

Table 12: **Generation task on MASI (Guo et al., 2025): quantization results across datasets.** FID ↓ is reported on MSD, LIDC, and COVID, with COVID runtime metrics. INT8 matches the FP16 baseline on FID (all gaps $\leq 0.2$) while improving throughput, latency, and memory, consistent with our core result that 8-bit quantization is near lossless. INT4 shows a clear FID degradation across datasets.

| Precision | FID ↓ | | | COVID runtime metrics | | |
|---|---|---|---|---|---|---|
| | MSD | LIDC | COVID | Throughput (samples/s) | Latency (s) | Memory (GB) |
| FP16 | 4.35 | 6.20 | 8.35 | 1.0 | 1.2 | 3.2 |
| INT8 | 4.42 | 6.35 | 8.52 | 1.8 | 0.7 | 1.6 |
| INT4 | 5.82 | 6.90 | 10.40 | - | - | 0.8 |

# I  INSIGHTS OF EFFICIENT MEDICAL MODEL ARCHITECTURES

**Efficient Model Architectures for Medical Vision.** The need of quantization-friendly medical vision architectures reveals a critical gap between architectural complexity and computational efficiency. Current state-of-the-art models, such as nnU-Net or MONAI frameworks, heavily rely on spatial convolution operations (e.g., 3D convolutions) to capture intricate anatomical structures in volumetric samples. While these operations perform well in spatial representation learning, they can introduce significant bottlenecks for applying quantization. Spatial convolutions often require a designed scale factor grouping across channels, tensors, or layers to maintain numerical stability during low-precision inference. This is a process that becomes increasingly error-prone with larger networks. In addition, the irregular memory access pattern inherent to 3D convolutions amplifies conversion overhead when converting models into optimized TensorRT or other engines, which will limit the practical gains of quantization.

On the other hand, transformer-based architectures, which have advantages for global context modeling, also show their challenges in quantization. Hybrid designs are still incorporating $3 \times 3 \times 3$ convolutional layers, such as those in SwinUNETR, and MedFormer inherit the quantization difficulties of both CNNs and attention mechanisms. For instance, the dynamic range of attention maps in ViTs often requires specialized quantization (adaptive stratification) to avoid information collapse during INT4 conversion. Meanwhile, hybrid conv layers disrupt the uniformity necessary for effective smoothing or singular value decomposition (SVD)-based quantization, which further complicates deployment. These architectural complexities underscore the need for a paradigm shift toward models explicitly designed for quantization efficiency, rather than relying on quantization onto existing architectures optimized solely for accuracy.

**Toward Quantization-Aware Architectural.** To address the above challenges, future medical vision architectures can target quantization-aware design principles without sacrificing spatial representation

robustness. One promising direction is the development of lightweight, hardware-aligned operators that can balance performance with low-precision robustness. For example, depthwise convolutions or Fourier-based spatial filters could reduce parameter redundancy while maintaining compatibility with INT8 optimizations. Similarly, attention mechanisms designed for medical imaging, such as sparse attention, could reduce the computational burden of full self-attention maps, which are notoriously sensitive to noise during quantization.

Another frontier is in designing medical models with emerging hardware. As platforms such as NVIDIA Blackwell and Rubin architectures provide support for sub-8-bit precision (e.g., FP6, INT4), medical AI models will need to evolve benchmarks that evaluate not only accuracy but also hardware-aware efficiency metrics such as energy-delay product (EDP) and memory utilization. For instance, architectures with regular computation, like hierarchical vision transformers with fixed patch sizes, may better exploit tensor core parallelism on later GPUs. Furthermore, generative models such as diffusion-based architectures (e.g., MAISI (Guo et al., 2025)) could benefit from quantization-friendly U-Net backbones that maintain high-resolution spatial modeling while enabling real-time synthesis on edge devices. By using quantization in architectural search pipelines and leveraging tools like model optimizer, researchers can flexibly identify optimal designs that can harmonize accuracy, efficiency, cost, and deployability.

Finally, the milestones to practical medical AI deployment hinge on closing the gap between simulation advancements and real-world constraints. Frameworks like MedQuanBench provide a critical foundation for evaluating quantization robustness. But the benchmark will require cross-disciplinary collaboration among researchers, hardware engineers, and clinicians to ensure that efficiency gains translate into real clinical workflows.

## J  POTENTIAL NEGATIVE SOCIETAL IMPACTS

The deployment of quantized medical imaging models may inadvertently amplify existing inequities in healthcare systems by prioritizing computational efficiency over diagnostic precision. Quantization-induced accuracy decreases could mismatch the effect of underrepresented populations if calibration datasets lack demographic diversity, which will lead to biased performance in critical tasks like tumor segmentation or anomaly detection. Furthermore, reliance on optimization frameworks risks creating technological bias and may lock resource-limited institutions into costly hardware. Overemphasis on benchmark metrics (e.g., Dice Score) without rigorous clinical validation might also obscure real-world trade-offs, such as delayed diagnoses or false negatives in time-sensitive scenarios. These risks highlight the ethical imperative to balance efficiency gains with equitable, transparent, and rigorously audited deployment practices to prevent harm to vulnerable patient populations.

## K  DECLARATION OF LLM TOOL USAGE

During the preparation of this manuscript, we used AI model for minor word selection, fixing grammar issues, and smoothing of the writing. The LLM tool was not used for generating original content, conducting data analysis, or formulating core scientific ideas. All conceptual development, experimentation, and interpretation were conducted independently without reliance on LLM tools. The other points involving the use of LLMs have already been highlighted in the paper.

## L  ETHICS STATEMENT

All authors of this work have read and commit to adhering to the ICLR Code of Ethics. We provide the potential impact and limitations on clinical applications below.

**Impact and Limitation on Clinical Application.** In real-world clinical settings, efficient and reliable AI inference is critical. Beyond edge devices, the quantization techniques have broader impacts on remote healthcare environments (e.g., cloud services, telesurgery) where infrastructure and communication capabilities are further limited. However, quantization methods inevitably involve trade-offs with accuracy, reliability, and robustness. Our benchmark results reveal the varying influences of quantization across different model components and layer choices. These insights can

enable practitioners to make informed decisions: whether to prioritize accuracy, maximize efficiency, or make a balance between the two, depends on specific clinical requirements and deployment.

## M REPRODUCIBILITY

To ensure reproducibility, we will provide a full open-source model and code shown in the manuscript.

