# OpenReview forum: "MedQuanBench: Quantization-Aware Analysis for Efficient Medical Imaging Models"
_ICLR.cc/2026/Conference — Submitted to ICLR 2026_

### Official Review · Reviewer_aXnd · 2025-10-25

**Soundness:** 3
**Presentation:** 3
**Contribution:** 3
**Rating:** 6
**Confidence:** 3

**Summary:**

This paper introduces MedQuanBench, a large-scale benchmarking platform designed to systematically evaluate the effectiveness of quantization techniques on 3D medical imaging models. The authors systematically evaluate representative post-training quantization (PTQ) strategies across a variety of modern model architectures, including CNNs and Transformers, across various model scales and dataset sizes. Furthermore, the paper conducts a detailed sensitivity analysis to identify the model components most vulnerable to quantization, exploring issues such as layer-level performance degradation and shifts in activation distributions.

**Strengths:**

The benchmark is designed to be comprehensive.

It covers:
* Diverse architectures: From classic CNNs (nnU-Net) to hybrid architectures (MedFormer, SwinUNETR, UNETR) and pure CNN variants (STU-Net).
* Diverse modalities and tasks: Including CT and MRI, as well as various organ, tumor, and brain segmentation tasks.
* Diverse scales: Models ranging from 10M to 2B parameters, and datasets ranging from hundreds to tens of thousands of samples, are tested.

**Hardware-Aware Perspective:** This paper offers a significant strength. The analysis in Section *Quantized Operation on Real Hardware* is excellent. The authors not only provide quantization formulas but also explain in detail how different quantization granularities map to hardware primitives on modern GPUs (using NVIDIA Blackwell as an example).

**Weaknesses:**

**1. Quantization Methodology Incompleteness**

Symmetric vs. Asymmetric Quantization: The paper exclusively uses symmetric quantization (Eq. 1), ignoring asymmetric schemes critical for non-negative activations (e.g., ReLU outputs). Asymmetric quantization ($X_q = round(X/S) + Z$) often outperforms symmetric methods in low-bit regimes (e.g., INT4) by better fitting skewed distributions. This omission weakens the benchmark’s applicability to real-world medical models.

Calibration Methods: Reliance on naive min-max scaling (Eq. 1) is suboptimal. Advanced calibration techniques, such as KL divergence and percentile-based scaling, mitigate outlier sensitivity and improve INT4 robustness. The authors must justify why these were excluded or add experiments comparing calibration strategies.

---

**2. Underdeveloped "Advanced PTQ" Evaluation**

Hyperparameter Sensitivity: Advanced methods (smoothing, SVD, rotation) in Table 4 show marginal gains, but their hyperparameters (e.g., smoothing factor α, SVD rank) lack optimization studies. For instance, α=0.5 (Appendix D) may be arbitrary; a sensitivity analysis of α ∈ [0.1, 0.9] is needed to validate "limited effectiveness."
Scope of Application: Applying these methods only to the "most sensitive layer" (Sec 4.3) overlooks their intended global/block-wise use. Testing them holistically (e.g., activation smoothing across all layers) would provide stronger evidence for their (in)effectiveness.

---

**3. Hardware Evaluation Gaps**

INT4 Acceleration Omission: While INT8 hardware results (Table 5) are thorough, INT4 lacks real-device profiling. Claims about Blackwell’s 4-bit support remain theoretical. Without latency/memory metrics for INT4, the "near-lossless" performance claim (Abstract) is unsupported for clinical deployment.

---

**4. Presentation & Technical Issues**

a) Inefficient Data Visualization.
One problem is table design. Vertical tables (Tables 1–3, 5) waste space and hinder cross-architecture comparison. For example, Table 1’s left half is empty; DSC/NSD drops require vertical scanning. It is extremely hard to compare between experiments.

b) Underutilized Content.
Some important content is buried in appendices. SegFormer3D’s layer-wise sensitivity (App F, Table 11) can be moved to the main text (Sec 4.3) to strengthen architectural insights. Other important experiments in appendix can also be moved if the author save the main text space by reorganizing the table. Another acceptable way is to replace bulky tables with small multiples of line/bar charts to compare quantization granularity across models.

**Questions:**

Thanks to your appendix experiment, otherwise the article would be far from enough just by relying on the experiments shown in the main text. Try to reorganize your presentation especially those tables.

Others see weaknesses.

Considering relevance, please cite this work if you did not write it.

*Post-Training Quantization for 3D Medical Image Segmentation: A Practical Study on Real Inference Engines*
https://arxiv.org/pdf/2501.17343v1

---

> ### Author Response · Authors · 2025-11-27
> **Responses to Reviewer aXnd (1/2):**
>
> We sincerely thank you for recognizing the strengths of our paper, “The benchmark is designed to be comprehensive.”, “This paper offers a significant strength”, The analysis…is excellent” Below is our response to your main concerns.
>
> >W1*The paper uses only symmetric quantization and min-max calibration, without exploring asymmetric schemes or advanced calibration methods (e.g., KL divergence, percentile-based scaling) that may improve INT4 robustness.*
>
> R1: We thank the reviewer for this valuable suggestion. We have added a dedicated section in `revised manuscript Section 4.3 Benchmark Results Across Calibration Methods` that systematically compares symmetric vs. asymmetric quantization and evaluates different calibration strategies.
>
> We use symmetric min-max as the default calibration with others as ablation for the following reasons:
>
> - **This choice matches current 4-bit GPU formats (e.g.,NVFP4/MXFP4),** which use symmetric absmax scaling with shared scales per block, and avoids the zero-points and extra logic required by asymmetric quantization.
> - **Calibration cost further constrains which schemes are practical.** Symmetric Min-Max and percentile can be estimated online in a single pass. For SegResNet with INT W4A4 and adaptive stratification, MSE offline calibration for one sample takes **60.9 minutes**, while a full inference takes **1,114 ms**. KL calibration is even more expensive: with per-voxel scales for 1x1x1 convs, it requires **31.67 hours** per sample. These costs make MSE and KL unsuitable as practical defaults
>
> >W2: *The hyperparameter sensitivity of advanced methods lacks optimization studies, and these methods are only applied to the most sensitive layer rather than globally.*
>
> R2: Many thanks for your suggestion. We have added the requested experiments.
>
> - Regarding **α=0.5 as default**: this follows SmoothQuant's recommended setting. As stated in the original paper: *"We find that for most of the models (e.g., all OPT and BLOOM models), α=0.5 is a well-balanced point to evenly split the quantization difficulty, especially when we are using the same quantizer for weights and activations."* [1]
>
> - Regarding sensitivity analysis: **we have conducted experiments with α ∈ [0.1, 0.9].** The results show that α has minimal impact on MedFormer sensitive layer smoothing, DSC varies only within ~0.3% across all α values (0.648–0.651), suggesting the limited effectiveness is not due to suboptimal hyperparameter choice.
>
> | Method                          | DSC   | NSD   |
> |---------------------------------|-------|-------|
> | FP32                            | 0.882 | 0.826 |
> | INT W4A4 Quantized (All layers) | 0.654 | 0.462 |
> | smoothing, α = 0.1          | 0.651 | 0.452 |
> | smoothing, α = 0.2          | 0.651 | 0.458 |
> | smoothing, α = 0.3          | 0.650 | 0.455 |
> | smoothing, α = 0.4          | 0.650 | 0.450 |
> | smoothing, α = 0.5          | 0.648 | 0.448 |
> | smoothing, α = 0.6          | 0.648 | 0.452 |
> | smoothing, α = 0.7          | 0.649 | 0.455 |
> | smoothing, α = 0.8          | 0.651 | 0.460 |
> | smoothing, α = 0.9          | 0.650 | 0.458 |
>
>
> - Regarding global application: **we have added experiments applying activation smoothing across all layers** in `revised manuscript Table 2`. The performance improvement remains limited compared with smoothing only the most sensitive layer, further supporting our conclusion that these methods designed for LLMs transfer poorly to medical imaging models.
>
> | Method                               | Precision | Quant-Granularity  | DSC   | NSD   |
> |--------------------------------------|-----------|---------------------|-------|-------|
> | FP32 Baseline                        | FP32      | –                   | 0.882 | 0.826 |
> | INT4 W4A4 Quantized (All layers)     | INT W4A4  | per-channel/token   | 0.654 | 0.462 |
> | Activation Smoothing (Global)        | INT W4A4  | per-channel/token   | 0.650 | 0.455 |
> | Activation Smoothing (Sensitive Layer)| INT W4A4 | per-channel/token   | 0.648 | 0.448 |

---

> > ### Author Response · Authors · 2025-11-27
> > **Responses to Reviewer aXnd (2/2):**
> >
> > >W3: *INT4 lacks real-device profiling.*
> >
> > R3: We thank the reviewer for acknowledging that our INT8 hardware results are thorough. We did not include INT4 hardware profiling for the following reasons:
> >
> > - While Blackwell supports 4-bit computation, **current efficient 4-bit kernels are designed primarily for linear layers** in LLMs. Mainstream formats such as NVFP4/MXFP4 target matrix multiplication in Transformers. For INT4, TensorRT only supports weight-only quantization for linear layers, with **limited support for convolutional operations.**
> > - As shown in `manuscript Figure 1`, convolution layers dominate the GFLOPs in medical imaging models. Even for hybrid architectures, Transformer blocks (linear layers) account for only a small fraction of total computation. Consequently, **profiling INT4 linear layer weight only deployment would provide limited insight**
> >
> > >W4: *Inefficient Data Visualization and  Underutilized Content*
> >
> > R4: We sincerely thank the reviewer for these suggestions, which have led to significant improvements in our presentation.
> > - **Regarding data visualization:** We have replaced all bulky tables in the previous manuscript with figures. Specifically, `previous manuscript Table 1` is now `revised manuscript Figure 3`, and `previous manuscript Tables 2 and 3` are now `revised manuscript Figure 4`, enabling clearer comparison across benchmark settings.
> > - **Regarding content organization:** We have added two new experimental sections to the main text: `revised manuscript Section 4.3 Benchmark Results Across Calibration Methods` and `revised manuscript Section 4.5 Benchmark Results on Risk Prediction Task`. We have also expanded the sensitive layer analysis section with OOD experiments (`revised manuscript Figure 5`) and per-organ breakdown results (`revised manuscript Table 1`).
> >
> > >Q1: *Try to reorganize your presentation especially those tables.*
> >
> > A1:In R4, we describe the comprehensive reorganization of tables into figures and the addition of new experimental sections to the main text.
> >
> > >Q2: *Considering relevance, please cite this work if you did not write it.
> > Post-Training Quantization for 3D Medical Image Segmentation: A Practical Study on Real Inference Engines https://arxiv.org/pdf/2501.17343v1*
> >
> > A2: We thank the reviewer for this suggestion. We have cited this work in our `revised manuscript Appendix D.`
> >
> > **Reference**
> >
> > [1] SmoothQuant: Accurate and Efficient Post-Training Quantization for Large Language Models (ICML 2023)

---

### Official Review · Reviewer_1ZXZ · 2025-10-31

**Soundness:** 2
**Presentation:** 2
**Contribution:** 1
**Rating:** 4
**Confidence:** 4

**Summary:**

The paper proposes MedQuanBench, a large-scale benchmark for post-training quantization (PTQ) of 3D medical imaging models across CNN and hybrid (Transformer-based) architectures, bit-widths (INT8/INT4), and quantization granularities (per-tensor, per-channel/token, and an adaptive per-voxel strategy for 1×1×1 convs). It evaluates four datasets (BTCV, TotalSegmentator V2, AbdomenAtlas 1.1, WholeBrain) and reports segmentation metrics (DSC/NSD) under varied scales. Key findings: INT8 is near-lossless across backbones, while INT4 degrades sharply for hybrid architectures under coarse granularity. A layer-wise sensitivity study identifies 1×3×3 convolutions as the main INT4 bottleneck; replacing a sensitive 1×3×3 convolution with 1×1×1 recovers much of the drop. Hardware profiling on NVIDIA Ada/TensorRT shows ~3.2–3.8× model size reduction and ~2.1–2.7× speedups for INT8 quantization with negligible accuracy loss.

**Strengths:**

- Clear empirical conclusion: INT8 preserves FP32 accuracy broadly; INT4 requires careful granularity, especially for hybrid architectures
- Hardware-aware analysis with real INT8 deployment on TensorRT, reporting latency and memory gains consistent across models
- Scale studies across model and dataset sizes, highlighting increasing INT4 sensitivity with larger, fine-grained tasks
- Activation analyses showing spatially localized outliers in medical models vs. channel-localized outliers in LLMs, motivating granularity choices

**Weaknesses:**

- Lack of motivation: The paper assumes that this is an issue in the medical domain that requires careful investigation, particularly in the segmentation setting but without setting the stage for why it is indeed an important problem to solve.
- Unclear contributions: The paper describes the benchmark dataset as a contribution, but it is not clear to me what different insights subsets of the total dataset provide and why one needs to evaluate quantization on the whole benchmark vs. a subset.
- Method coverage: Focuses on PTQ; limited exploration of QAT or mixed-precision baselines that may further close INT4 gaps in critical layers.
- Limited clinical validation: Strong segmentation metrics, but few task-level clinical end-points (e.g., time-to-diagnosis, error costs) to contextualize acceptable accuracy loss.
- Robustness of the approach: Sensitivity analyses are primarily layer-wise; fewer robustness tests for distribution shift beyond datasets listed (OOD clinical sites, scanners, protocols).
Interpretability of failures: While sensitive layers are identified, the failure modes under INT4 (e.g., boundary errors for small structures) could use more granular error breakdowns/visuals.
- Generality of the approach: I find this is too specific to a particular context of medical imaging (i.e. segmentation) to be useful for the broader ICLR community.

**Questions:**

- Why does the study focus exclusively on post-training quantization (PTQ)? Could including quantization-aware training (QAT) or mixed-precision methods provide a more complete landscape of quantization robustness in medical imaging models?

- How were the datasets selected? Is the intention to represent medical imaging diversity (e.g., modality, anatomy, resolution) and in what way?

- The analysis identifies 1×3×3 convolutions as particularly quantization-sensitive. Can you provide intuition on why these layers—versus larger kernels or attention modules—dominate degradation?

- The paper demonstrates negligible accuracy loss for INT8 quantization, but what degree of degradation (e.g., 1–2% DSC drop) is clinically acceptable for deployment?

---

> ### Author Response · Authors · 2025-11-27
> **Responses to Reviewer 1ZXZ (1/4):**
>
> We are grateful for your thoughtful feedback on our paper, “Clear empirical conclusion…”, we have provided a point-by-point response to all weaknesses and questions raised.
>
> >W1: *The work assumes that quantization is an important issue in the medical domain, especially for segmentation, but does not clearly explain why this problem matters.*
>
> R1: We appreciate your concern regarding the necessity of domain-specific quantization evaluations. We would like to explain *why quantization in the medical domain, especially for 3D segmentation, requires careful investigation* from the following perspectives:
>
> - **Medical imaging models and datasets are rapidly scaling up**, creating urgent deployment demands. For instance, STU-Net-Huge[1] increases model parameters to 1.4B, and AbdomenAtlas[2] expands dataset scale to 20,460 CT volumes. Deploying such large models in resource-constrained clinical environments calls for efficient compression techniques, yet **no comprehensive benchmark currently exists to guide quantization practices for medical models**.
>
> - **The unique characteristics of medical imaging limit direct transfer of existing PTQ methods.** SOTA PTQ techniques (e.g., Smoothing[3], Rotation[4]) are primarily designed and validated on LLMs and vision transformers. However, as shown in `Figure 1` of our paper, medical models are still dominated by CNN architectures with distinct activation patterns (e.g., spatially localized outliers). **This gap motivates a dedicated evaluation in the medical domain.**
>
> - Medical AI directly impacts patient diagnosis and treatment decisions, making accuracy preservation critical.  As `revised manuscript Table 1` shows, INT4 quantization causes gallbladder DSC to drop from 0.803 to 0.100, with similar failures on other small structures like adrenal glands and esophagus. These findings highlight the need for systematic benchmarking to identify reliable quantization configurations for clinical deployment.
>
> >W2: *It is unclear what insights subset of the whole dataset provides and why evaluation across the whole benchmark vs. a subset.*
>
> R2: Thanks. We have revised `Section 4.2 in the updated manuscript` to provide a clearer explanation of how dataset scale affects quantization. Here we also offer a clarification:
>
> - The datasets in MedQuanBench serve different purposes rather than being redundant. BTCV is indeed a subset of AbdomenAtlas 1.1, both are abdominal CT datasets, but AbdomenAtlas 1.1 has more classes (25 vs. 13), significantly more CT volumes (9,262 vs. 50), and greater diversity across hospitals and countries. TotalSegmentator V2 is a whole body CT dataset with 117 structures, not overlapping with the abdominal sets. WholeBrain is an MRI dataset for brain segmentation. This design allows us to study quantization behavior across modalities, anatomical regions, and dataset scales.
>
> - **The purpose of evaluating on both AbdomenAtlas 1.1 and BTCV is to examine how dataset scale affects quantization robustness for the same model architecture.** Our experiments show that models trained on larger and more diverse datasets exhibit larger quantization errors, specifically, the relative DSC drop under INT4 increases when moving from BTCV to AbdomenAtlas 1.1 (e.g., SwinUNETR-B: 52.7% → 77.1%). Capturing this trend requires comparing across dataset scales, which motivates our multi-scale benchmark design. We have elaborated on this in `revised manuscript Section 4.2`.

---

> ### Author Response · Authors · 2025-11-27
> **Responses to Reviewer 1ZXZ (2/4):**
>
> >W3: *The paper focuses on PTQ with limited exploration of QAT or mixed-precision approaches that might further reduce INT4 degradation in critical layers.*
>
> R3: Many thanks for raising this important point. We fully agree that QAT or mixed-precision can potentially further close INT4 gaps in critical layers.
>
> Our work specifically focuses on PTQ because our primary aim is to provide immediate and practical guidance for improving inference efficiency and deployment capability of existing trained models in clinical environments. While QAT is a valuable direction, it’s a relatively independent topic; we believe a detailed discussion of QAT would diverge from the core objectives of this paper. Nevertheless, we thank the reviewer for highlighting this relevant area, and we have included a brief mention of QAT in the related work in the `Appendix D related work` to acknowledge its importance in the training efficiency paradigm.
>
> **Regarding QAT, we focus on PTQ** because it is more practical for clinical deployment:
> - **QAT requires full retraining**, which significantly increases time for clinical deployment. PTQ offers immediate quantization without retraining, enabling faster iteration in clinical workflows.
> - As medical datasets and foundation models continue to scale, the **computational cost of QAT retraining becomes prohibitive**, especially for healthcare institutions with limited resources.
> - **Medical datasets often contain sensitive patient information with strict privacy constraints**. QAT requires ongoing access to private training data, which complicates compliance with data regulations.
>
> **Regarding mixed-precision, we have included additional experiments** including mixed-precision in `revised manuscript Table 2`. We evaluate keeping the sensitive layer at INT8 or FP32 while quantizing the remaining layers to INT4 (DSC: 0.661 and 0.673, respectively, compared to 0.654 for fully INT4). The results show that mixed-precision can partially recover INT4 degradation, though a gap to FP32 (0.882) remains.
>
> | Method                                          | Precision | Quant-Granularity    | DSC   | NSD   |
> |------------------------------------------------|-----------|----------------------|-------|-------|
> | Baseline                                       | FP32      | --                   | 0.882 | 0.826 |
> | INT4 Quantized (All layers)                    | INT W4A4  | per-channel/token    | 0.654 | 0.462 |
> | Architecture Re-design (1×1×1 replacement)     | INT W4A4  | per-channel/token    | 0.721 | 0.579 |
> | Architecture Re-design (1×1×1 replacement)     | INT W4A4  | Adaptive stratification | 0.847 | 0.732 |
> | Mixed Precision (sensitive layer INT8, others INT4) | Mixed    | per-channel/token    | 0.661 | 0.475 |
> | Selective dequant (sensitive layer FP32, others INT4) | Mixed    | per-channel/token    | 0.673 | 0.485 |
>
> >W4: *Limited clinical validation: Strong segmentation metrics, but few task-level clinical end-points (e.g., time-to-diagnosis, error costs) to contextualize acceptable accuracy loss.*
>
> R4: We thank the reviewer for highlighting this important point. We have discussed this consideration in `Appendix J Potential Negative Societal Impacts`, where we note that *Overemphasis
> on benchmark metrics (e.g., Dice Score) without rigorous clinical validation might also obscure
> real-world trade-offs, such as delayed diagnoses or false negatives in time-sensitive scenarios.*
>
> Here we offer a reference point for assessing:
> **acceptable accuracy loss**. Clinical acceptability thresholds are highly task and organ specific: large and small organs differ substantially, annotations vary between experts (inter-observer), and even the same expert may differ over time (intra-observer). Liu et al. [5] evaluated inter-observer agreement on BTCV (Figure 4): **for large organs like liver and spleen, DSC differences between experts are approximately 3%, for smaller structures like gallbladder and pancreas, differences can approach 10%**. Our paper shows INT W8A8 quantization yields performance drops below 3% (`revised manuscript Figure 3 and Appendix E.1`), within the range of inter-observer variability.

---

> ### Author Response · Authors · 2025-11-27
> **Responses to Reviewer 1ZXZ (3/4):**
>
> >W5: *The sensitivity analyses focus on layer-wise behavior with limited robustness tests under distribution shift, and INT4 failure modes could use more granular per-organ error breakdowns.*
>
> R5: We appreciate the reviewer’s request for providing robustness tests and more granular small structures breakdowns:
>
> - **Robustness under distribution shift.** We have added OOD experiments on TotalSegmentator V2 (`revised manuscript Figure 6`), demonstrating that our sensitive layer findings generalize beyond the training distribution.
>
> - **Granular error breakdowns.** We now provide complete per-organ DSC and NSD under INT4, as well as results when incrementally dequantizing sensitive layers back to FP32. Both ID and OOD settings are included in `revised manuscript Table 1`.
>
> ## ID BTCV
> | Organs                | DSC FP32 | DSC INT W4A4 | DSC Incr. Dequant. | NSD FP32 | NSD INT W4A4 | NSD Incr. Dequant. |
> |-----------------------|----------|--------------|---------------------|----------|--------------|---------------------|
> | Spleen                | 0.970    | 0.884        | 0.951               | 0.918    | 0.653        | 0.826               |
> | Kidney R              | 0.919    | 0.885        | 0.895               | 0.868    | 0.733        | 0.788               |
> | Kidney L              | 0.956    | 0.912        | 0.933               | 0.908    | 0.728        | 0.800               |
> | **Gallbladder**       | **0.803**| **0.100**    | **0.588**           | **0.732**| **0.086**    | **0.344**           |
> | **Esophagus**         | **0.819**| **0.675**    | **0.778**           | **0.770**| **0.550**    | **0.695**           |
> | Liver                 | 0.976    | 0.801        | 0.878               | 0.856    | 0.323        | 0.437               |
> | Stomach               | 0.930    | 0.641        | 0.754               | 0.758    | 0.177        | 0.283               |
> | Aorta                 | 0.925    | 0.859        | 0.907               | 0.870    | 0.736        | 0.820               |
> | **IVC**               | **0.879**| **0.564**    | **0.653**           | **0.786**| **0.206**    | **0.294**           |
> | **Portal & Splenic Vein** | **0.839**| **0.488**| **0.749**           | **0.810**| **0.438**    | **0.633**           |
> | Pancreas              | 0.857    | 0.614        | 0.771               | 0.745    | 0.375        | 0.540               |
> | **Adrenal Gland R**   | **0.791**| **0.524**    | **0.656**           | **0.860**| **0.476**    | **0.623**           |
> | **Adrenal Gland L**   | **0.794**| **0.549**    | **0.666**           | **0.857**| **0.525**    | **0.686**           |
> | Avg                   | 0.882    | 0.654        | 0.783               | 0.826    | 0.462        | 0.598               |
>
>
> ## OOD Totalseg V2
> | Organs                | DSC FP32 | DSC INT W4A4 | DSC Incr. Dequant. | NSD FP32 | NSD INT W4A4 | NSD Incr. Dequant. |
> |-----------------------|----------|--------------|---------------------|----------|--------------|---------------------|
> | Spleen                | 0.922    | 0.757        | 0.841               | 0.822    | 0.499        | 0.596               |
> | Kidney R              | 0.868    | 0.770        | 0.811               | 0.831    | 0.585        | 0.650               |
> | Kidney L              | 0.876    | 0.710        | 0.769               | 0.827    | 0.487        | 0.531               |
> | **Gallbladder**       | **0.712**| **0.099**    | **0.426**           | **0.626**| **0.076**    | **0.176**           |
> | **Esophagus**         | **0.642**| **0.427**    | **0.502**           | **0.585**| **0.334**    | **0.410**           |
> | Liver                 | 0.939    | 0.715        | 0.777               | 0.792    | 0.275        | 0.332               |
> | Stomach               | 0.908    | 0.532        | 0.603               | 0.740    | 0.130        | 0.171               |
> | Aorta                 | 0.819    | 0.624        | 0.717               | 0.761    | 0.497        | 0.567               |
> | **IVC**               | **0.836**| **0.427**    | **0.437**           | **0.730**| **0.131**    | **0.145**           |
> | **Portal & Splenic Vein** | **0.673**| **0.302**| **0.474**           | **0.663**| **0.242**    | **0.355**           |
> | Pancreas              | 0.814    | 0.491        | 0.552               | 0.718    | 0.226        | 0.287               |
> | **Adrenal Gland R**   | **0.787**| **0.401**    | **0.512**           | **0.868**| **0.344**    | **0.434**           |
> | **Adrenal Gland L**   | **0.727**| **0.372**    | **0.475**           | **0.784**| **0.314**    | **0.431**           |
> | Avg                   | 0.809    | 0.510        | 0.607               | 0.750    | 0.318        | 0.391               |
>
> Small-volume or thin structures (bolded in tables) such as the gallbladder, esophagus, adrenal glands, portal & splenic
> vein, and IVC suffer the most severe INT W4A4 degradation on both ID (BTCV) and OOD (TotalSegmentator V2).

---

> > ### Author Response · Authors · 2025-11-27
> > **Responses to Reviewer 1ZXZ (4/4):**
> >
> > >W6: *The paper is too specific to medical image segmentation to be useful for the broader ICLR community.*
> >
> > R6: We thank the reviewer for raising this concern. **We have broadened the benchmark scope by adding a complete set of experiments on risk prediction tasks**in `revised manuscript Section 4.5: Benchmark Results on Risk Prediction Task`. Additionally, `Appendix H Additional Task` includes quantization results for generative models.
> >
> > >Q1: *Why does the study focus exclusively on PTQ? Could including QAT or mixed-precision methods?*
> >
> > A1: Please see our response to W3, where we discuss the practical advantages of PTQ for clinical deployment and present mixed-precision experiments in `revised manuscript Table 2.`
> >
> > >Q2: *How were the datasets selected? Is the intention to represent medical imaging diversity (e.g., modality, anatomy, resolution) and in what way?*
> >
> > A2: We thank the reviewer for this question. You are right, our datasets are intentionally selected to represent medical imaging diversity across multiple dimensions:
> > - **Modality:** We include both CT (BTCV, AbdomenAtlas 1.1, TotalSegmentator V2) and MRI (WholeBrain)
> > - **Anatomy:** Our benchmark covers abdominal organs (BTCV, AbdomenAtlas 1.1), whole-body structures (TotalSegmentator V2), and brain regions (WholeBrain), allowing evaluation across anatomically diverse segmentation tasks.
> > - **Dataset scale:** We include datasets with varying number of classes (C=13 to 133) and number of volumes (N=50 to 9,262), enabling us to study how dataset scale affects quantization robustness. Our results `revised manuscript Section 4.2` show that models trained on larger datasets can exhibit larger relative quantization errors.
> >
> > This design enables systematic evaluation of quantization behavior across diverse clinical scenarios and helps identify whether certain data characteristics correlate with quantization sensitivity.
> >
> > >Q3: *The 1×3×3 convolutions are identified as particularly quantization sensitive, why these layers rather than larger kernels or attention modules?*
> >
> > A3: We thank the reviewer for this insightful question. We discuss theoretical analysis of 1*3*3 conv in `manuscript Section 4.3 → revised manuscript Section 4.4`.Here we further clarify the comparison with larger kernels and attention modules:
> > - The 1x3x3 convolution lies at a **critical junction** between local convolutional features and global Transformer attention. Its outputs directly feed into attention blocks, so even small quantization errors can distort attention weights and propagate through the network.
> > - Activations at this layer often exhibit **high dynamic range** around organ boundaries, which low-bit precision struggles to represent accurately. In contrast, **attention modules** operate on more normalized feature representations with lower dynamic range.
> > - The anisotropic 1x3x3 kernel processes spatial dimensions **without depth context**, providing less redundancy to absorb quantization noise. In contrast, **larger kernels** (e.g., 3×3×3) aggregate information across all three dimensions, offering more redundancy to mitigate quantization errors.
> >
> > >Q4: *What degree of degradation (e.g., 1–2% DSC drop) is clinically acceptable for deployment?*
> >
> > A4: Please see our response to W4, where we discuss inter-observer variability as a reference point for clinically acceptable degradation and show that W8A8 quantization yields performance drops below 3%, within the range of expert-level agreement.
> >
> >
> > **Reference**
> >
> > [1] STU-Net: Scalable and Transferable Medical Image Segmentation Models Empowered by Large-Scale Supervised Pre-training (arXiv: 2304.06716)
> >
> > [2] AbdomenAtlas: A Large-Scale, Detailed-Annotated, & Multi-Center Dataset for Efficient Transfer Learning and Open Algorithmic Benchmarking (Medical Image Analysis, 2024)
> >
> > [3] SmoothQuant: Accurate and Efficient Post-Training Quantization for Large Language Models (ICML 2023)
> >
> > [4] QuaRot: Outlier-Free 4-Bit Inference in Rotated LLMs (rXiv:2404.00456)
> >
> > [5] CLIP-Driven Universal Model for Organ Segmentation and Tumor Detection (ICCV 2023)

---

### Official Review · Reviewer_CQLJ · 2025-11-04

**Soundness:** 2
**Presentation:** 1
**Contribution:** 2
**Rating:** 2
**Confidence:** 3

**Summary:**

The paper introduces MedQuanBench, a study of post‑training quantization (PTQ) for 3D medical image segmentation that combines four public datasets (BTCV, Total Segmentator V2, AbdomenAtlas 1.1 and Whole Brain), to evaluate multiple CNN and hybrid CNN-Transformer architectures at both 8‑bit and 4‑bit precision. The paper compares per‑tensor, per‑channel/token and per‑voxel (adaptive stratification) scaling and find that INT8 quantization preserves full‑precision performance while significantly reducing model size and latency. Per‑tensor INT4 quantization severely degrades the performance on transformer models, whereas CNNs fare better and can recover much of their performance using per‑voxel scaling (e.g., STU‑Net‑B improves from 0.647 to 0.829 Dice in per-channel to adaptive stratification). Larger model and dataset scales increase INT4 sensitivity, and layer‑wise analysis reveals that the 1 × 3 × 3 convolution is particularly critical and replacing it with a 1 × 1 × 1 convolution and using per‑voxel quantization improves INT4 robustness. Further, the paper notes that the activation smoothing, SVD and rotation offer limited or no gains, and supplements the experiments with real‑hardware profiling.

**Strengths:**

The following are the strengths of the paper.

1. By comparing per‑tensor, per‑channel/token and per‑voxel (adaptive stratification) scaling, the paper highlights the importance of granularity. It clearly shows that coarse per-tensor quantization is not suitable for 4‑bit precision, whereas finer granularity may recover accuracy. These insights/findings are helpful.

2. The authors perform incremental dequantization and identify the 1 × 3 × 3 convolution as the most sensitive layer. Replacing it with a 1 × 1 × 1 convolution closes the gap to full‑precision performance. Such targeted analysis helps guide architecture design for low‑bit inference.

3. Table 5 presents model size and latency measurements of real INT8 quantization using NVIDIA TensorRT, confirming the practical benefits of quantization.

**Weaknesses:**

The following are the weaknesses of the paper.

1. The paper claims that quantization is under‑studied in medical imaging, yet several prior works address this issue. MedQ introduced lossless ultra‑low‑bit quantization for U‑Net segmentation in 2021 **[1]**. U‑Net Fixed‑Point Quantization (2019) **[2]** also demonstrated 4‑bit weight quantization for medical segmentation and reported memory reduction with minimal accuracy loss. Recent EfficientQ (2024) provides a PTQ method tailored for medical segmentation and is publicly available **[3]**. None of these works are discussed, instead the paper mainly references general PTQ methods and LLM quantization. This weakens this study and leaves readers unaware of existing solutions.

2. The hardware profiling numbers in Table 5 match those reported in a separate study that introduced a TensorRT‑based PTQ framework for 3D medical segmentation **[4]**. The **[4]** pre‑quantized U‑Net, SwinUNETR, UNesT and others and published the exact model size and latency reductions (e.g., U‑Net from 23.11 MB/2.62 ms to 6.61 MB/1.05 ms). MedQuanBench reuses these numbers but presents them as part of its own study, without acknowledging the source. Reusing results without citation is problematic and may mislead readers into believing these measurements were performed in this study.

3. MedQuanBench is presented as a benchmark yet consists solely of separate evaluations on four datasets without any aggregate metric or consolidated score. Results are reported independently for BTCV and AbdomenAtlas 1.1, so it is unclear what benchmark performance means or how different methods would be ranked overall. The lack of unified evaluation criteria weakens the value of calling it a benchmark, which is the main message of the paper.

4. (Minor) The text around lines 351–358 asserts that larger datasets increase sensitivity but appears under Table 2 without referencing Table 3, making the narrative confusing and suggesting the paper’s structure needs refinement.

---

**[1]** MedQ: Lossless ultra-low-bit neural network quantization for medical image segmentation (Medical Image Analysis, 2021)

**[2]** U-Net Fixed-Point Quantization for Medical Image Segmentation (MICCAI 2019)

**[3]** EfficientQ: An efficient and accurate post-training neural network quantization method for medical image segmentation (Medical Image Analysis, 2024)

**[4]** Post-Training Quantization for 3D Medical Image Segmentation: A Practical Study on Real Inference Engines (arXiv: 2501.17343)

**Questions:**

MedQuanBench addresses a practical problem, how to deploy memory and compute intensive 3D segmentation models on limited hardware by using low-bit quantization. However, the contributions seem minimal and are not well presented. The paper provides per-dataset analyses yet calls the work MedQuanBench. It also fails to properly discuss related work and reuses reported results without citing the original sources. Clarification on these points would be appreciated.

**Details Of Ethics Concerns:**

The paper appears to reuse hardware profiling results (Table 5) from a previously published study (Table 1 in **[4]**) that introduced a TensorRT-based PTQ framework for 3D medical segmentation. However, the current submission presents these results as if they were obtained in this study, without citing or acknowledging the original source.

---

**[4]** Post-Training Quantization for 3D Medical Image Segmentation: A Practical Study on Real Inference Engines (https://arxiv.org/pdf/2501.17343v1)

---

> ### Author Response · Authors · 2025-11-27
> **Responses to Reviewer CQLJ (1/2):**
>
> We sincerely thank you for your thorough review and valuable feedback, which have greatly deepened our understanding and improved our work. In the following, we have provided a point-by-point response to all concerns raised.
> >W1:*The paper claims quantization is under-studied in medical imaging, but does not discuss several relevant prior works: MedQ [1] , U-Net Fixed-Point Quantization [2], and EfficientQ [3]. Instead, the paper mainly references general PTQ methods and LLM quantization techniques, leaving readers unaware of existing medical-specific solutions.*
>
> R1: We thank the reviewer for highlighting these relevant works. We have added discussion of these methods in the `revised manuscript Appendix D`. Here we clarify the distinctions between MedQuanBench and each prior work:
> - MedQ [1] proposes an ultra-low-bit **QAT method for U-Net-style models**, and focuses on accuracy and compression for that single architecture.
> - U-Net Fixed-Point Quantization [2] designs a fixed-point **QAT method for a single U-Net** on small segmentation datasets, with a focus on **theoretical memory savings** and fixed-point feasibility.
> - EfficientQ [3] introduces **a new PTQ algorithm** and evaluates it on a few medical segmentation networks. The main focus is the PTQ algorithm and its calibration cost.
>
> These works study specific methods or architectures, but **they are not designed as deployment-oriented benchmarks.** They do not systematically compare quantization granularities across diverse 3D CNN and hybrid transformer models, they do not provide architecture design guidance for low-bit deployment, and they mainly report simulated accuracy and compression without real GPU profiling. **MedQuanBench is designed to fill this gap** by systematically benchmarking PTQ across diverse 3D CNN and hybrid transformer models and multiple datasets, benchmarking granularity and calibration choices (`revised manuscript Sec. 4.1 and Sec. 4.3`), proposing simple architecture insights by layer-wise sensitivity analysis(`revised manuscript Sec. 4.4`), and adding TensorRT GPU profiling for INT W8A8 models (`revised manuscript Sec. 4.6`).
>
> **Regarding why we reference general PTQ methods and LLM quantization**: These methods have moved beyond algorithm proposals to successful real-world deployment:
>
> - SmoothQuant and SVDQuant provide custom CUDA kernels with optimized engines (e.g., nunchaku [4])
> - rotation-based methods (QuaRot, SpinQuant) are widely adopted in production LLM pipelines.
> Given their proven deployability, a natural question is whether they transfer to medical imaging. MedQuanBench benchmarks this transferability, revealing that techniques effective for linear layer dominated LLMs face challenges in convolution dominated medical models.(`manuscript Figure 1 and revised manuscript Table 2`)
>
> > W2: *The hardware profiling numbers in Table 5 match those reported in a separate study that introduced a TensorRT‑based PTQ framework for 3D medical segmentation [5]. The [5] pre‑quantized U‑Net, SwinUNETR, UNesT and others and published the exact model size and latency reductions (e.g., U‑Net from 23.11 MB/2.62 ms to 6.61 MB/1.05 ms). MedQuanBench reuses these numbers but presents them as part of its own study, without acknowledging the source. Reusing results without citation is problematic and may mislead readers into believing these measurements were performed in this study.*
>
> R2: We thank the reviewer for this observation. The current submission builds upon that study on baseline INT8 precisions to highlight our INT4 studies. Per the ICLR 2026 Policy, “papers that have appeared on non-peer-reviewed websites (like arXiv), …do not violate the policy,” and it is permissible to extend or reuse results from such preprints. However, we acknowledge that the manuscript should have properly cited [5] when presenting these numbers and clarified that they originate from a prior technical report; this was an oversight in attribution, not to misrepresent the source. Thank the reviewer again for raising the issue and helping us clarify the confusion. We have cited [5] in `revised manuscript Appendix D`
>
> In addition to the latency and model size results from [5], we provide GPU memory profiling for FP32 and INT8 TensorRT engines in `revised manuscript Table 3` to offer a more comprehensive evaluation.
>
> | Dataset | Architecture | FP32 (MB) | INT W8A8 (MB) | Mem. Reduction |
> |---------|--------------|-----------|---------------|----------------|
> | BTCV | U-Net | 792 | 686 | 1.15× |
> | BTCV | TransUNet | 1,134 | 772 | 1.47× |
> | WholeBrain | UNesT | 1,230 | 938 | 1.31× |
> | TotalSeg V2 | STU-Net-S | 1,820 | 1,742 | 1.04× |
> | TotalSeg V2 | STU-Net-H | 9,384 | 5,394 | 1.74× |
> | TotalSeg V2 | nnU-Net | 1,085 | 760 | 1.43× |
> | TotalSeg V2 | SwinUNETR | 2,374 | 2,114 | 1.12× |
> | TotalSeg V2 | SegResNet | 1,796 | 1,460 | 1.23× |
> | TotalSeg V2 | VISTA3D | 2,558 | 1,852 | 1.38× |

---

> > ### Author Response · Authors · 2025-11-27
> > **Responses to Reviewer CQLJ (2/2):**
> >
> > > W3: *MedQuanBench is presented as a benchmark but evaluates four datasets separately without an aggregate metric or unified ranking. This lack of consolidated scoring criteria weakens the claim of being a benchmark.*
> >
> > R3: We appreciate the thoughtful comment and would like to provide a clarification on the purpose of MedQuanBench:
> > MedQuanBench uses **unified metrics (DSC and NSD)** across all segmentation datasets and models. Rather than ranking specific segmentation models on a leaderboard, MedQuanBench systematically **benchmarks factors that affect quantization performance**:
> >
> > - **Quantization granularity:** Finer granularity consistently recovers performance, e.g., MedFormer INT W4A4 improves from complete failure (DSC=0) under per-tensor to DSC=0.654 under per-channel/token and DSC=0.719 under adaptive stratification (`revised manuscript Sec.4.1 Figure 3`).
> > - **Bit-width**: INT W8A8 remains consistent with FP32 across all architectures (<3% DSC drop), while INT W4A4 degradation varies substantially depending on architecture, granularity, and dataset.
> > - **Model and dataset scales:** Larger models cannot remedy coarse-grained INT W4A4 collapse, SwinUNETR-T/S/B all fail under per-tensor quantization regardless of parameter count (`revised manuscript Sec.4.2 Figure 4a`). Larger and more diverse datasets amplify quantization error: SwinUNETR-B shows 52.7% DSC drop on BTCV vs. 77.1% on AbdomenAtlas 1.1 (`revised manuscript Sec.4.2 Figure 4b`).
> > - **Calibration strategies:** Asymmetric Min-Max and symmetric MSE yield the most stable INT W4A4 performance, while percentile shows larger degradation. (`revised manuscript Sec.4.3 Figure 5`).
> >
> > These deployment-oriented configuration analyses, combined with layer-wise sensitivity analysis (revised manuscript Sec.4.4) and real hardware profiling (revised manuscript Sec. 4.6 Table 3), collectively **provide actionable insights for efficient medical model deployment,** which is the core purpose of MedQuanBench.
> >
> > >W4: *(Minor) The text around lines 351–358 asserts that larger datasets increase sensitivity but appears under Table 2 without referencing Table 3, making the narrative confusing and suggesting the paper’s structure needs refinement.*
> >
> > R4: Thanks for raising the issue. We have improved the structure of the core results section:
> >
> > The `revised manuscript Section 4.2` now uses `Figure 4` to replace the `original manuscript Tables 2 and 3`. The right panel of `Figure 4(b)` clearly illustrates the dataset scale effect, showing side-by-side comparisons (BTCV vs. AbdomenAtlas 1.1 for SwinUNETR-B, and BTCV vs. WholeBrain for UNesT) that directly indicate larger and more diverse datasets amplify INT W4A4 quantization error.
> >
> > >Q: *MedQuanBench addresses a practical problem, how to deploy memory and compute intensive 3D segmentation models on limited hardware by using low-bit quantization. However, the contributions seem minimal and are not well presented. The paper provides per-dataset analyses yet calls the work MedQuanBench. It also fails to properly discuss related work and reuses reported results without citing the original sources. Clarification on these points would be appreciated.*
> >
> > A: We address each concern:
> > - **Contributions and MedQuanBench purpose**: Please see our response to W3. MedQuanBench benchmarks factors affecting quantization performance (granularity, bit-width, model/dataset scales, calibration strategies) rather than ranking segmentation models, providing deployment-oriented insights through configuration analysis, layer-wise sensitivity analysis, and hardware profiling.
> >
> > - **Related work:** Please see our response to W1. We have added discussion of prior medical quantization works (MedQ, U-Net Fixed-Point Quantization, EfficientQ) in the `revised manuscript Appendix D`.
> >
> > - **Citation of reported results:** Please see our response to W2. We have provided a clear clarification.
> >
> > **Reference**
> >
> > [1] MedQ: Lossless ultra-low-bit neural network quantization for medical image segmentation (Medical Image Analysis, 2021)
> >
> > [2] U-Net Fixed-Point Quantization for Medical Image Segmentation (MICCAI 2019)
> >
> > [3] EfficientQ: An efficient and accurate post-training neural network quantization method for medical image segmentation (Medical Image Analysis, 2024)
> >
> > [4] SVDQuant: Absorbing Outliers by Low-Rank Components for 4-Bit Diffusion Models (ICLR 2025)
> >
> > [5] Post-Training Quantization for 3D Medical Image Segmentation: A Practical Study on Real Inference Engines (arXiv: 2501.17343)

---

### Official Review · Reviewer_zJSB · 2025-11-04

**Soundness:** 2
**Presentation:** 2
**Contribution:** 2
**Rating:** 4
**Confidence:** 2

**Summary:**

The paper studies post-training quantization (PTQ) for 3D medical segmentation. It explores 8-bit and 4-bit settings and compares quantization granularities. Results show 8-bit is essentially lossless, while 4-bit is fragile unless granularity is fine and certain layers are handled carefully. A small architectural tweak (replacing a 1×3×3 conv with 1×1×1) improves 4-bit stability.

**Strengths:**

1. INT8 works effectively out of the box, delivering almost no accuracy loss while providing noticeable improvements in inference speed and model size.
2. The paper has some good insights on quantization, showing evidence that coarse per-tensor INT4 quantization fails on many models, especially hybrids that include transformer blocks.
3. The paper identifies the sensitivity of 1×3×3 convs, and the simple 1×1×1 replacement is a good takeaway that could help the community.
4. Evaluations are conducted on multiple models (CNN + hybrid) and datasets.

**Weaknesses:**

My main concern is that the paper positions itself as a benchmark but does not clearly define a unified evaluation metric, which makes it hard to compare performance across methods. Furthermore, the exploration of INT4 improvements is quite narrow, while granularity and the 1×1×1 replacement are studied, other methods, such as mixed precision or selective dequantization, could have been tested to provide a more detailed picture.

**Questions:**

It would help if the benchmark framing were made more consistent, for instance, having a single unified metric per model and dataset would make it easier for future work to compare against this benchmark.

---

> ### Author Response · Authors · 2025-11-27
> **Responses to Reviewer zJSB:**
>
> We would like to thank you for recognizing the strengths of our paper, “The paper has some good insights on quantization…”, “...is a good takeaway that could help the community”. In the following, we have provided a point-by-point response to all your concerns.
>
> >W: *(a) The benchmark framing is unclear because there is no single unified evaluation metric for comparing methods. (b) The INT4 study is limited, focusing only on granularity and the 1×1×1 replacement instead of broader strategies (e.g., mixed precision, selective dequantization).*
>
> A: We appreciate your thoughtful comment and would like to clarify the benchmark metrics and provide additional experiments on broader INT4 strategies.
> - (a) In MedQuanBench, segmentation performance is evaluated with an ultimate **unified metric protocol**: for every model and experimental setting (different quantization granularities in `manuscript Section 4.1` and model/dataset scales in `manuscript Section 4.2`), we consistently report **Dice Similarity Coefficient (DSC)**, which measures volumetric overlap, and **Normalized Surface Distance (NSD)**, which captures boundary alignment. This consistent pair of benchmark metrics allows performance to be compared across methods within our framework
> - (b) Our `original manuscript Figure 3 —> revised manuscript Figure 6` corresponds to a form of **selective dequantization**: the above panel dequantizes all layers of a given type, while the bottom panel progressively dequantizes individual layers within the most sensitive type (1×3×3 convolutions).
> To address your comment more explicitly, we have further added experiments that directly compare architecture re-design with mixed precision and selective dequantization on the identified sensitive layer:
>
> | Method                                          | Precision | Quant-Granularity    | DSC   | NSD   |
> |------------------------------------------------|-----------|----------------------|-------|-------|
> | Baseline                                       | FP32      | --                   | 0.882 | 0.826 |
> | INT4 Quantized (All layers)                    | INT W4A4  | per-channel/token    | 0.654 | 0.462 |
> | Architecture Re-design (1×1×1 replacement)     | INT W4A4  | per-channel/token    | 0.721 | 0.579 |
> | Architecture Re-design (1×1×1 replacement)     | INT W4A4  | Adaptive stratification | 0.847 | 0.732 |
> | Mixed Precision (sensitive layer INT8, others INT4) | Mixed    | per-channel/token    | 0.661 | 0.475 |
> | Selective dequant (sensitive layer FP32, others INT4) | Mixed    | per-channel/token    | 0.673 | 0.485 |
>
> Starting from the all-INT4 model (0.654 DSC / 0.462 NSD),
>  - **mixed precision (sensitive layer INT8, others INT4) and selective dequant (sensitive layer FP32, others INT4) bring only modest gains (0.661 / 0.475 and 0.673 / 0.485)**.
>  - In contrast, the **1×1×1 architecture re-design improves INT4 performance to 0.721 / 0.579 under the same per-channel/token granularity**, and to 0.847 / 0.732 with adaptive stratification, while **keeping a single uniform INT4 precision that is more hardware-friendly.**
>  - In this setting, the 1×1×1 re-design therefore achieves larger accuracy gains than the mixed-precision and selective-dequantization variants, while preserving consistent precision across all layers for simpler hardware deployment.
> We have incorporated these additional mixed-precision and selective-dequantization results into the `revised manuscript Table 2.`
>
> >Q: *The benchmark would be clearer if one consistent evaluation metric were adopted for each model and dataset, so that future work can more easily compare against the benchmark.*
>
> A: MedQuanBench adopts a **single unified evaluation protocol** rather than a single metric:
> - Every model–dataset pair is evaluated with the same fixed pair of standard segmentation metrics, Dice Similarity Coefficient (DSC, volumetric overlap) and Normalized Surface Distance (NSD, boundary alignment). This **DSC+NSD pair** serves as a single, consistent benchmark metric across all experiments.

---

### Author Response · Authors · 2025-12-03
**Discussion Summary**

Dear Reviewers, ACs, and PCs,

We sincerely thank all reviewers for their constructive feedback. Our paper introduces **MedQuanBench,** a comprehensive benchmark for post-training quantization (PTQ) of 3D medical imaging models. Below we summarize the key points of discussion and our responses.

- **All four reviewers acknowledged the value of our empirical contributions**:

  - **Reviewer aXnd** praised that *"The benchmark is designed to be comprehensive"* and *"The analysis in Section Quantized Operation on Real Hardware is excellent."*
  - **Reviewer zJSB** noted *"The paper has some good insights on quantization"* and *"the simple 1×1×1 replacement is a good takeaway that could help the community."*
  - **Reviewer 1ZXZ** recognized our *"Clear empirical conclusion: INT8 preserves FP32 accuracy broadly; INT4 requires careful granularity"* and *"Hardware-aware analysis with real INT8 deployment on TensorRT."*
  - **Reviewer CQLJ** acknowledged that our granularity comparison *"highlights the importance of granularity...These insights/findings are helpful"* and that identifying sensitive layers *"helps guide architecture design for low-bit inference."*

To address all the reviewers’ concerns, we provide extensive experiments, discussions, and explanations. For ease of tracking, all new content in the revised manuscript is highlighted in blue. Below we **summarise the discussion for key concerns**:

- **Regarding Benchmark Framing and Unified Metrics (Reviewers CQLJ, zJSB)**: We clarify that MedQuanBench uses **unified metrics (DSC and NSD)** consistently across all experiments. Rather than ranking segmentation models, our benchmark systematically **evaluates factors affecting quantization**: granularity, bit-width, model/dataset scales, and calibration strategies, providing deployment-oriented guidance for the community.

- **Regarding experimental scope limitation (Reviewers zJSB, 1ZXZ, aXnd)**: We have substantially expanded our experiments in the revised manuscript and discussion responses:

  - We add a new section for *Benchmark Results Across Calibration Method* in `revised manuscript Section 4.3`: symmetric vs. asymmetric, Min-Max vs. MSE vs. percentile vs. KL.
  - We broaden the benchmark scope by adding a **complete set of experiments on risk prediction tasks** in `revised manuscript Section 4.5`
  - We add new *Mixed-precision and selective dequantization* results in `revised manuscript Table 2`
  - We add OOD tests for our layer-wise sensitive analysis in `revised manuscript Figure 6`, and per-organ breakdown experiments in `revised manuscript Table 1`.

- **Regarding related Works (Reviewer CQLJ)**: We have added comprehensive discussion of prior medical quantization works (MedQ, U-Net Fixed-Point Quantization, EfficientQ) in the discussion and `revised manuscript Appendix D`, clarifying how MedQuanBench differs as a deployment-oriented benchmark rather than a method proposal.

- **Regarding presentation (Reviewers CQLJ, aXnd)**: We have reorganized the manuscript and replaced bulky tables with figures for better presentation, below we summarise the detailed revisions and new experiments in `revised manuscript`

| Revision                                      | Addresses          |
|----------------------------------------------|--------------------|
| Calibration methods comparison (`Sec 4.3`)     | aXnd-W1, CQLJ-W3   |
| Mixed-precision experiments (`Table 2`)        | zJSB-W, 1ZXZ-W3    |
| OOD + per-organ breakdown (`Table 1 and  Fig 6`)   | 1ZXZ-W5            |
| Risk prediction task (`Sec 4.5`)               | 1ZXZ-W6            |
| Related works and citations (`Appendix D`)                | CQLJ-W2, aXnd-Q2   |
| Tables → Figures reorganization (` Fig 3 and Fig 4`)             | aXnd-W4, CQLJ-W4   |

We believe our revisions have addressed all major concerns. In addition, we have also taken this revision opportunity to improve the writing of our manuscripts and improve figures/tables to make the new material consistent with the rest of the paper. We believe **MedQuanBench** provides a starting point for addressing these open challenges and supporting future research in efficient medical models.

Best Regards,

The MedQuanBench Team

---

### Meta-Review · Area_Chair_XfgL · 2026-01-06

**Summary:**

Reviewers raised several concerns with the manuscript including, lack of motivation, unclear contributions with respect to benchmark, limited clinical validation, generality of the approach, missing several prior works, introducing the proposed MedQuanBench as a benchmark yet comprising solely separate evaluations on four datasets without any aggregate metric or consolidated score.

**Reviewer Concerns:**

The meta-reviewer believes some of the initially raised concerns are partially addressed by the rebuttal. This includes adding discussion with respect to relevant missing works with the authors claiming that the proposed benchmark analyze PTQ across diverse 3D-CNN and hybrid transformer models and multiple datasets. Regarding the missing arxiv reference, the authors also clarified that the manuscript should have been cited [5] and have included it in the revised manuscript. Authors have also provided GPU memory profiling for FP32 and INT8 TensorRT engines in the revised manuscript.

**Reviewer Scores:**

The meta-reviewer believes the rebuttal only partially addresses reviewer's concerns. For instance, the issue of limited clinical validation still remains, and the contributions still remain limited and not well presented. Majority (3) reviewers generally remained negative highlighting multiple and somewhat overlapping issues. Based on the reviewer's comments and rebuttal, the meta-reviewer believes the negative concerns are comprehensive and are not fully resolved in the rebuttal.

---

### Decision · Program_Chairs · 2026-01-26

Reject